# Transcriptomic encoding of sensorimotor transformation in the midbrain

Zhiyong Xie[1†], Mengdi Wang[2,3†], Zeyuan Liu[2,3†], Congping Shang[4†], Changjiang Zhang[2,3], Le Sun[5], Huating Gu[1], Gengxin Ran[2,3], Qing Pei[1], Qiang Ma[2,3], Meizhu Huang[4], Junjing Zhang[6], Rui Lin[1], Youtong Zhou[1], Jiyao Zhang[6], Miao Zhao[1], Minmin Luo[1,7], Qian Wu[6]*, Peng Cao[1,8]*, Xiaoqun Wang[2,3,4,5,7,9]*

[1]National Institute of Biological Sciences, Beijing, China; [2]State Key Laboratory of Brain and Cognitive Science, CAS Center for Excellence in Brain Science and Intelligence Technology (Shanghai), Institute of Biophysics, Chinese Academy of Sciences, Beijing, China; [3]University of Chinese Academy of Sciences, Beijing, China; [4]Bioland Laboratory (Guangzhou Regenerative Medicine and Health Guangdong Laboratory), Guangzhou, China; [5]Beijing Institute for Brain Disorders, Capital Medical University, Beijing, China; [6]State Key Laboratory of Cognitive Neuroscience and Learning, IDG/McGovern Institute for Brain Research, Beijing Normal University, Beijing, China; [7]Chinese Institute for Brain Research, Beijing, China; [8]Tsinghua Institute of Multidisciplinary Biomedical Research, Tsinghua University, Beijing, China; [9]Beijing Advanced Innovation Center for Big Data-Based Precision Medicine, Beihang University & Capital Medical University, Beijing, China

*For correspondence:
qianwu@bnu.edu.cn (QW);
caopeng@nibs.ac.cn (PC);
xiaoqunwang@ibp.ac.cn (XW)

†These authors contributed equally to this work

**Abstract** Sensorimotor transformation, a process that converts sensory stimuli into motor actions, is critical for the brain to initiate behaviors. Although the circuitry involved in sensorimotor transformation has been well delineated, the molecular logic behind this process remains poorly understood. Here, we performed high-throughput and circuit-specific single-cell transcriptomic analyses of neurons in the superior colliculus (SC), a midbrain structure implicated in early sensorimotor transformation. We found that SC neurons in distinct laminae expressed discrete marker genes. Of particular interest, *Cbln2* and *Pitx2* were key markers that define glutamatergic projection neurons in the optic nerve (Op) and intermediate gray (InG) layers, respectively. The Cbln2+ neurons responded to visual stimuli mimicking cruising predators, while the Pitx2+ neurons encoded prey-derived vibrissal tactile cues. By forming distinct input and output connections with other brain areas, these neuronal subtypes independently mediated behaviors of predator avoidance and prey capture. Our results reveal that, in the midbrain, sensorimotor transformation for different behaviors may be performed by separate circuit modules that are molecularly defined by distinct transcriptomic codes.

## Introduction

Sensorimotor transformation is a fundamental process in which the brain converts sensory information into motor command (*Crochet et al., 2019*; *Franklin and Wolpert, 2011*; *Pouget and Snyder, 2000*). The critical role of this process in sensory-guided behaviors has been demonstrated in diverse animal models, including fish (*Bianco and Engert, 2015*; *Chen et al., 2018*; *Helmbrecht et al., 2018*), rodents (*Felsen and Mainen, 2008*; *Huda et al., 2020*; *Mayrhofer et al., 2019*; *Oliveira and Yonehara, 2018*; *Wang et al., 2020a*), and primates (*Buneo et al., 2002*; *Cavanaugh et al., 2012*;

*Sparks, 1986*). Although the brain circuits and computational models of sensorimotor transformation have been intensively studied, the molecular and genetic logic behind this process remains elusive.

Single-cell RNA-sequencing (scRNA-seq) and single-nucleus RNA-sequencing (snRNA-seq) are powerful approaches to identify the genes expressed in individual cells (*Liu et al., 2020*; *Shapiro et al., 2013*; *Tang et al., 2009*; *Zhong et al., 2020*; *Zhong et al., 2018*), enabling us to understand the cellular diversity and gene expression profiles of a specific brain region (*Economo et al., 2018*; *Saunders et al., 2018*; *Zeisel et al., 2018*). Moreover, by combining scRNA-seq with tools for circuit analysis, one should be able to link the transcriptomic heterogeneity to other characteristics of neurons such as their electrophysiological properties (*Földy et al., 2016*), spatial distribution (*Eng et al., 2019*; *Moffitt et al., 2018*; *Shah et al., 2016*), neuronal activity (*Hrvatin et al., 2018*; *Liu et al., 2020*; *Wu et al., 2017*), and projection specificity (*Tasic et al., 2018*). Thus, scRNA-seq may provide an opportunity to explore the molecular and genetic logic of sensorimotor transformation.

In the mammalian brain, the superior colliculus (SC) is a midbrain structure for early sensorimotor transformation (*Basso and May, 2017*; *Cang et al., 2018*). The superficial layers of the SC, including the superficial gray (SuG) layer and the optic nerve (Op) layer, are involved in visual information processing (*De Franceschi and Solomon, 2018*; *Gale and Murphy, 2014*; *Wang et al., 2010*). The deep layers of the SC, including the intermediate layer and deep layer, participate in processing of tactile and auditory information (*Cohen et al., 2008*; *Dräger and Hubel, 1975*). The deep layers of the SC control eye movement (*Sparks, 1986*; *Wang et al., 2015*), head movement (*Isa and Sasaki, 2002*; *Wilson et al., 2018*), and locomotion (*Felsen and Mainen, 2008*). From a neuroethological perspective, the sensorimotor transformations that occur in the SC enable it to orchestrate distinct behavioral actions in predator avoidance and prey capture (*Dean et al., 1989*; *Oliveira and Yonehara, 2018*). However, how different neuronal subtypes participate in these survival behaviors and the molecular features of these neurons remain unknown.

In the present study, by performing high-throughput and circuit-specific single-cell transcriptomic analyses of cells in the SC, we systematically studied the molecular markers of SC neurons, sensory response properties, input-output connectivity, and their behavioral relevance. We found that Cbln2 + and Pitx2+ SC neurons form part of two distinct sets of circuit modules for sensorimotor transformation related to behaviors of predator avoidance and prey capture. Our data suggest that sensorimotor transformation for different behaviors may be performed by separate circuit modules that are molecularly defined by distinct transcriptomic codes.

## Results

### A census of SC cell types using snRNA-seq

To understand the cell diversity of the SC, snRNA-seq of mouse SC was performed using the 10× Genomics Chromium Platform. From two experimental replicates, each containing six superior colliculi, 14,892 single-cell gene expression profiles were collected (*Figure 1A*, *Figure 1—figure supplement 1A–B*; *Supplementary file 1*). In total, we found nine major types of cells identified by the expression of classic marker genes; these were excitatory neurons, inhibitory neurons, astrocytes, oligodendrocyte progenitor cells (OPCs), oligodendrocytes, microglia, endothelial cells, ciliated cells, and meningeal cells (*Figure 1B*, *Figure 1—figure supplement 1C*). To further investigate neural diversity, we divided the excitatory and inhibitory neurons into 9 and 10 subclusters, respectively, each of which displayed a distinctive transcriptomic profile (*Figure 1A,C–D*; *Supplementary file 2*). The differentially expressed genes (DEGs) expressed by the cells in these subsets indicate that subclusters In-5 and In-10 are Calb1+ and Reln+ interneurons, respectively (*Figure 1D*).

Since the SC possesses a layered structure with a variety of circuit connections (*Doubell et al., 2003*), we next asked whether the subsets of neurons we identified are located in specific layers. To answer this question, we developed a method of spatial classification of mRNA expression data (SPACED) (https://github.com/xiaoqunwang-lab/SPACED [*Wang, 2021a*]; also see the Materials and methods section for details) through which we were able to assign a location score to each neural subset by analyzing RNA in situ hybridization images of the top DEGs in each subset from Allen Brain Atlas (https://mouse.brain-map.org) (*Figure 1E–F*, *Figure 1—figure supplement 1D*). Using this method, a specificity score and the statistical significance were calculated for each

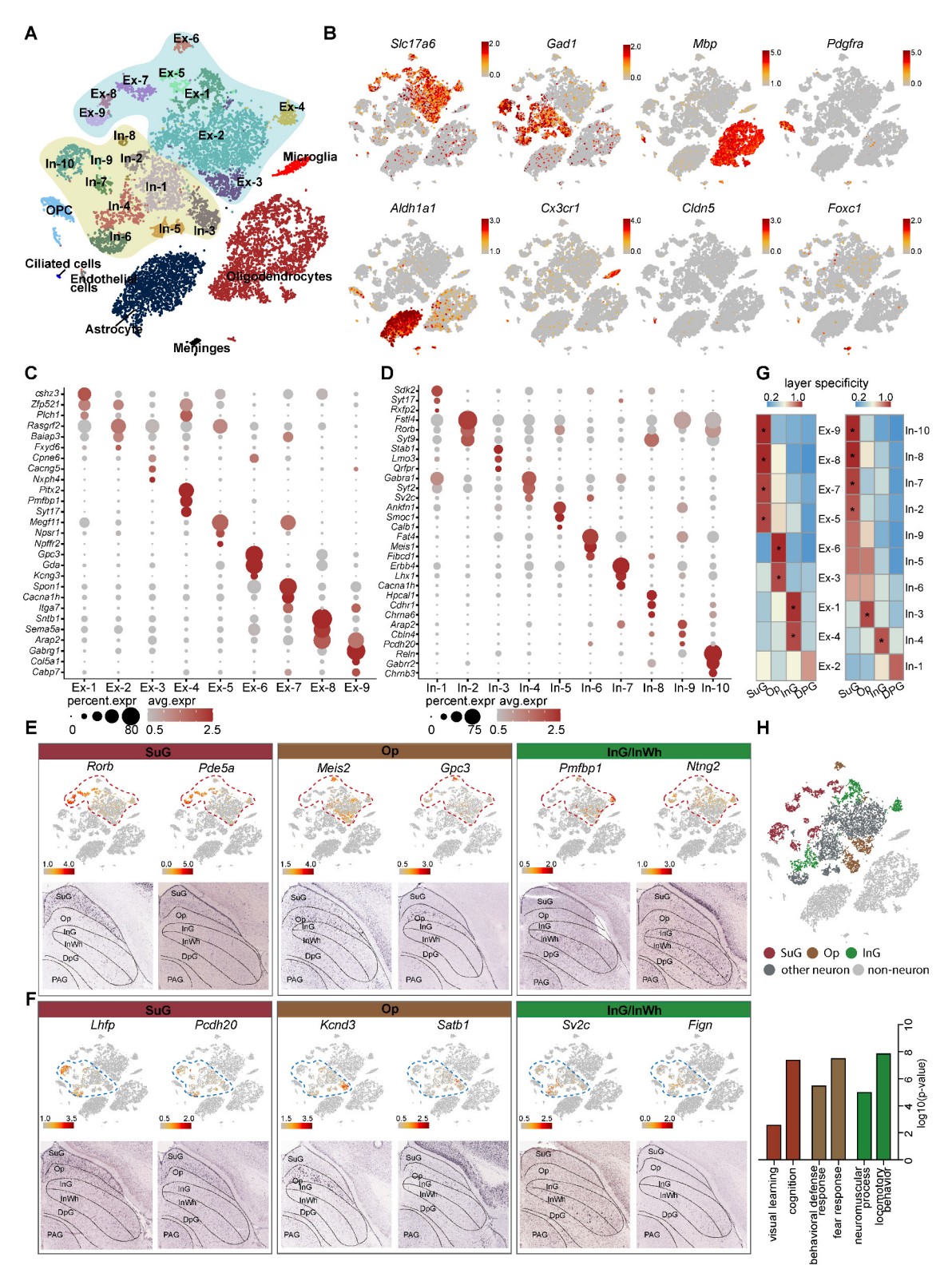

**Figure 1.** Identification and characterization of cell types and spatial heterogeneity of mouse superior colliculus (SC) neurons. (**A**) Unbiased clustering of single-nucleus RNA-sequencing (snRNA-seq) data of mouse SC cells. Each dot represents an individual cell. The cells were grouped into 26 clusters, and the cell types were annotated according to the expression of known marker genes. (**B**) t-Distributed stochastic neighbor embedding (t-SNE) showing the known markers of major cell types (excitatory neurons, inhibitory neurons, oligodendrocytes, oligodendrocyte progenitor cells (OPCs),

*Figure 1 continued on next page*

*Figure 1 continued*

astrocytes, microglia, endothelial cells, ciliated cells and meningeal cells) in the mouse SC. The scale bar indicates the relative gene expression level (gray, low; red, high). (C, D) Dot plots showing the differentially expressed genes (DEGs) among 9 excitatory neuron subclusters (C) and 10 inhibitory neuron subclusters (D). (E, F) Spatial expression of the top DEGs of excitatory neuron subclusters (E) and inhibitory neuron subclusters (F). Upper panel: gene expression levels projected onto the two-dimensional t-SNE and colored according to relative gene expression level (gray, low; red, high). Red dashed line, excitatory neuron subclusters; blue dashed line, inhibitory neuron subclusters. Lower panel: in situ hybridization staining of mouse SC for the identified excitatory neuron layer markers (from the Allen Brain Atlas). (G) Heatmap showing the computed layer specificity score for each excitatory neuron subcluster (left) and each inhibitory neuron subcluster (right). Statistical analyses were performed by ANOVA (*p<0.05). (H) SC layer information annotation of neurons. Upper panel: cells colored by layer information as indicated by the legend on the bottom. Lower panel: gene ontology enrichment analysis of layer-annotated SC neurons.

The online version of this article includes the following figure supplement(s) for figure 1:

**Figure supplement 1.** Quality of single-nucleus RNA-sequencing (snRNA-seq) metrics and the spatial distribution of neurons.
**Figure supplement 2.** Comparison of neuronal subtypes and spatial mapping results.

subset, and it was found that the subsets of excitatory neurons and inhibitory neurons exhibited distinctive specificities for different layers of the SC (*Figure 1E–G*). Cells of subsets Ex-5/7/8/9 and In-2/7/8/10 were assigned to the SuG matter layer, while cells in the Ex-3/6 and In-3 subsets localized in the Op layer. In addition, Ex-1/4 and In-4 cells exhibited high spatial scores for the intermediate gray and white (InG/InWh) layers (*Figure 1E–G*, *Figure 1—figure supplement 1D*) (p < 0.05). Although several neural subsets also showed relatively high spatial scores for certain layers, the data for those subsets did not meet the criteria for statistical significance, indicating that some neurons might be located in multiple layers. To determine the reliability of the spatial assignments, we compared the mapping results obtained from SPACED with those from previously published method (*Zeisel et al., 2018*; *Figure 1—figure supplement 2A–B*). We extracted excitatory neurons and inhibitory neurons from Zeisel's work with the assigned regional identities of SC and integrated those cells with SC neurons from this study (*Figure 1—figure supplement 2C–F*). Co-embedding of neuronal subtypes showed identical regional identities, indicating that SPACED produced essentially consistent results with Zeisel's method. In addition, SPACED methods offered more detailed spatial information of SC neural subtypes (*Figure 1—figure supplement 2D,F*). We next analyzed the gene ontology (GO) enrichment of the DEGs of cells that we assigned to different layers (*Figure 1H*). Intriguingly, the GO terms suggest that predicted SuG layer cells may play roles in visual learning and cognition, consistent with previous findings that cells in this layer receive signals from the retina and the visual cortex (*Sparks, 1986*). The GO analysis also indicated that cells that are predicted in the Op layer may be involved in defense and fear responses, while cells that are predicted in the InG layer may play roles in locomotor behavior (*Figure 1H*). These data suggest that neurons with diverse and distinctive transcriptomic profiles may be located in different layers of the SC.

## Electrophysiological properties of LPTN- and ZI-projecting SC neurons

Several populations of glutamatergic neurons in the SC show distinctive projection patterns (*Dean et al., 1989*). To accurately visualize layer-specific neuronal projection patterns from the SC to downstream brain regions, we utilized a dual-AAV expression system that enables sparse labeling of individual neurons with neuronal subtype specificity (*Lin et al., 2018*). We injected a mixture of AAV (AAV-TRE-DIO-Flpo and AAV-TRE-fDIO-GFP-IRES-tTA) into the SC of vGlut2-IRES-Cre mice (*Figure 2A*) and performed morphological connectivity reconstruction by image tracing of individual cells (M-CRITIC) (https://github.com/xiaoqunwang-lab/M-CRITIC; *Wang, 2021b*, copy archived at swh:1:rev:f7eab14cfe2e13c807f923349556b85a3ee31c61; also see the Materials and methods section for details). The complete morphological structure was reconstructed from multiple consecutive two-photon image-tracing stacks after alignment and was registered to the Allen Common Coordinate Framework (CCF) (*Figure 2B*, *Figure 2—video 1*). One neuron with a cell body in the Op layer and dendritic ramifications in the SuG and Op layers extended its axon in the lateral posterior thalamic nucleus (LPTN). Another neuron showed dendrites restricted to InG/InWh layer and a branched axon reaching the zona incerta (ZI) (*Figure 2C*). To map how LPTN-projecting neurons are distributed in the SC, we injected AAV2-retro-DIO-EGFP into the LPTN of vGlut2-IRES-Cre mice (*Figure 2D*). Retrogradely labeled EGFP+ SC neurons were predominantly localized within the Op

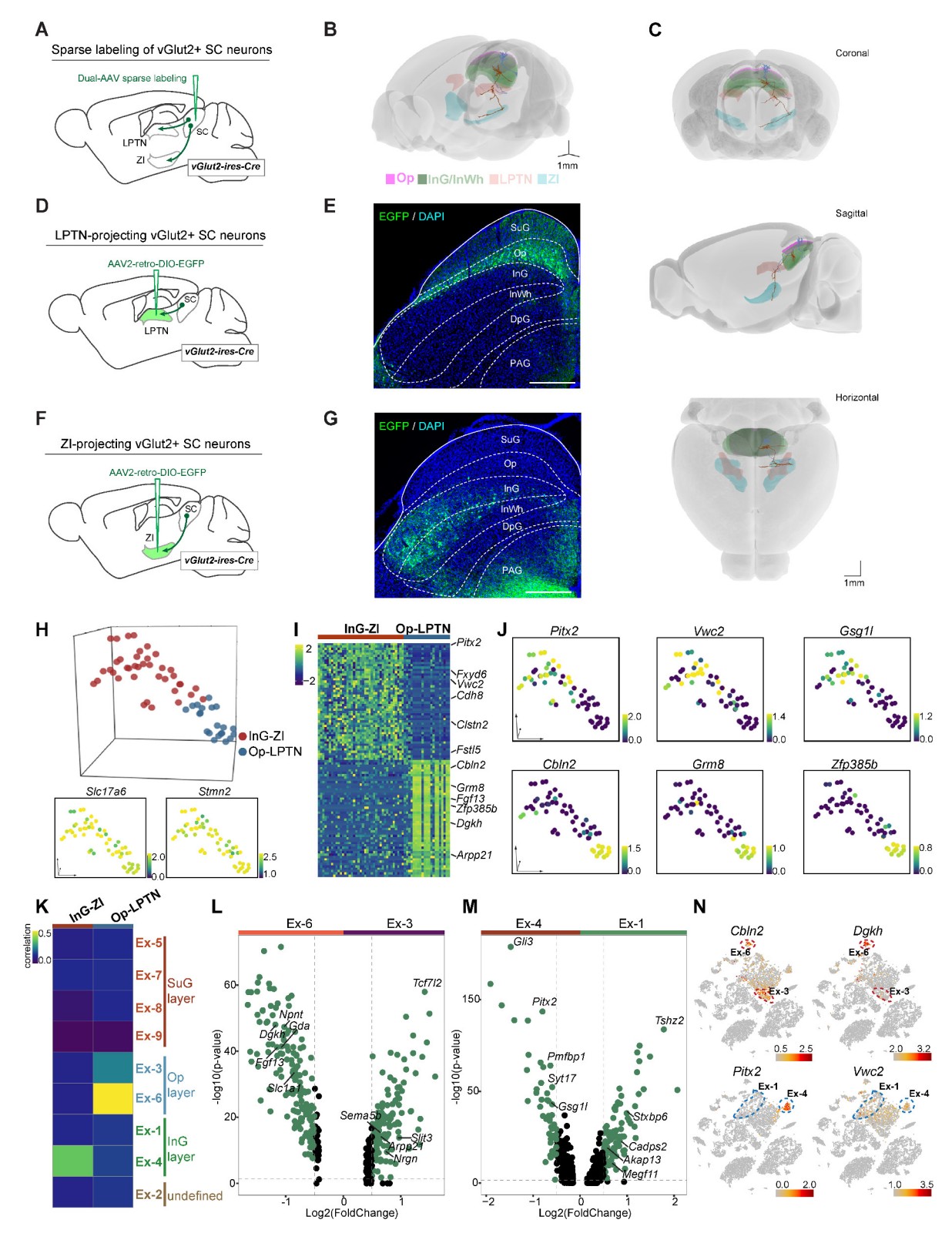

**Figure 2.** Projection-based analyses of single-cell gene expression profiles. (**A**) Schematic diagram showing injection of AAV mixture into the superior colliculus (SC) of vGlut2-IRES-Cre mice for sparse labeling of glutamatergic SC neuron projections. (**B**) Two reconstructed neurons (blue, cells projected from the optic nerve [Op] to the lateral posterior thalamic nucleus [LPTN]; brown, cells projected from the intermediate gray [InG] to the zona incerta [ZI]) were registered to the mouse brain regions (Op, magenta; LPTN, orange; InG/intermediate white (InWh), green; ZI, cyan). (**C**) Coronal, *Figure 2 continued on next page*

*Figure 2 continued*

sagittal, and horizontal views of reconstructed neurons. Scale bar, 1 mm. (**D**) Schematic diagram showing injection of AAV2-retro-DIO-EGFP into the LPTN of vGlut2-IRES-Cre mice for labeling of LPTN-projecting glutamatergic SC neurons. (**E**) Sample micrograph showing the distribution of LPTN-projecting glutamatergic SC neurons labeled by EGFP. (**F**) Schematic diagram showing injection of AAV2-retro-DIO-EGFP into the ZI of vGlut2-IRES-Cre mice for labeling of ZI-projecting glutamatergic SC neurons. (**G**) Sample micrograph showing the distribution of ZI-projecting glutamatergic SC neurons labeled by EGFP. (**H**) Three-dimensional (3D) t-distributed stochastic neighbor embedding (t-SNE) plot showing SC cells sequenced by patch-seq. Upper panel: 3D t-SNE displaying the distribution of Op-LPTN and InG-ZI projection neurons. Cells are colored according to their cell projection identities (Op-LPTN, blue; InG-ZI, red). Lower panel: expression profiles of classic markers *Slc17a6* and *Stmn2* for VGlut-expressing neurons were projected onto the 3D t-SNE. The scale bar indicates the relative gene expression level (blue, low; yellow, high). (**I**) Heatmap showing the differentially expressed genes of ZI- and LPTN-projecting SC neurons. The scale bar indicates the relative gene expression level. (**J**) Expression of genes enriched in LPTN-projecting and ZI-projecting neurons visualized as a t-SNE plot (blue, low; yellow, high). (**K**) Transcriptional correlation between LPTN-projecting and ZI-projecting neurons (patch-seq) and excitatory neuron subtypes (high-throughput single-nucleus RNA-sequencing [snRNA-seq]). The scale bar indicates the correlation coefficient (blue, low; yellow, high). (**L, M**) Volcano plot showing the differentially expressed genes in excitatory neuron subtypes Ex-3 and Ex-6 (**L**) and excitatory neuron subtypes Ex-1 and Ex-4 (**M**). Each dot represents a gene. Significantly upregulated genes are shown in green. (**N**) t-SNE plot visualizing the expression of differentially expressed genes in LPTN-projecting (top) and ZI-projecting (bottom) neurons in the same layout used in *Figure 1A*. The scale bar indicates the relative gene expression level (gray, low; red, high).

The online version of this article includes the following video and figure supplement(s) for figure 2:

**Figure supplement 1.** Electrophysiological properties and the expression of differentially expressed genes (DEGs) between lateral posterior thalamic nucleus (LPTN)- and zona incerta (ZI)-projecting superior colliculus (SC) neurons.

**Figure 2—video 1.** 3D reconstruction of layer-specific neuron projection patterns from SC to downstream brain regions by in vivo sparse-labeling strategy and M-CRITIC, related to *Figure 2B–C*.

https://elifesciences.org/articles/69825#fig2video1

layer (*Figure 2E*). With a similar strategy, we labeled ZI-projecting SC neurons with EGFP; these neurons are distributed in the InG/InWh and deep gray (DpG) layers of the SC (*Figure 2F–G*).

To compare the electrophysiological properties of these two neuronal populations, we performed whole-cell current-clamp recordings from LPTN-projecting and ZI-projecting SC neurons in acute SC slices (*Figure 2—figure supplement 1A*). These two populations of SC projection neurons did not show significant differences in resting membrane potential (*Figure 2—figure supplement 1B*) or firing threshold (*Figure 2—figure supplement 1C*). The number of action potentials fired by LPTN-projecting and ZI-projecting neurons in response to membrane depolarization also did not show a significant difference (*Figure 2—figure supplement 1D–E*). These data suggest that LPTN-projecting and ZI-projecting neurons are similar in their electrophysiological properties and that they cannot be distinguished using traditional electrophysiological measurements.

## Projection-specific single-cell transcriptomic analysis

Next, we prepared acute SC slices and collected EGFP+ cells from the SC for patch-seq experiments (*Cadwell et al., 2016*; *Liu et al., 2020*). In total, 78 cells were collected; 60 of these cells, including 21 LPTN-projecting neurons from the Op layer and 39 ZI-projecting neurons from the InG layer, passed the quality control test, with a median number of 7746 genes expressed per cell (*Figure 2— figure supplement 1F–G*; *Supplementary file 3*). Examination of classic markers indicated that these cells were vGlut-expressing neurons and the neurons with different projections were clustered separately (*Figure 2H*), indicating that neurons with the same circuit connections may have similar innate gene expression profiles. We then further analyzed the DEGs of these two neuronal populations. The InG layer neurons projecting to ZI with soma in the SC highly expressed *Pitx2*, *Vwc2*, *Gsg1l*, *Clstn2*, and other genes, while LPTN-projecting Op layer neurons highly expressed *Cbln2*, *Grm8*, *Zfp385b*, *Dgkh*, and other genes (*Figure 2I–J*, *Figure 2—figure supplement 1H–I*; *Supplementary file 4*). On examination of the high-throughput snRNA-seq data, we found that two excitatory neuron subsets (Ex-3, 6) were assigned to the Op layer and two subsets (Ex-1, 4) belonged to the InG layer (*Figure 1G*). Transcriptomic correlation analysis indicated that Op-LPTN neurons and InG-ZI neurons identified through patch-seq were similar to cells of the Ex-6 and Ex-4 subsets, respectively, in terms of their cellular gene expression profiles (*Figure 2K*, *Figure 2—figure supplement 1J*). We next analyzed the DEGs of two subsets of cells from the Op layer (Ex-3 vs. Ex6) and the InG layer (Ex-1 vs. Ex-4) (*Figure 2L–M*; *Supplementary file 5*). We found that some projection-specific genes were restricted to one subset of cells (*Figure 2L–N*, *Figure 2—figure supplement 1K*). For example, as a marker gene of Op-LPTN projection neurons, *Cbln2* was expressed in

both Ex-3 and Ex-6 cells, while *Dgkh* was only highly expressed in Ex-6 cells (*Figure 2L,N*). Among genes that were specifically expressed in InG-ZI projection neurons, *Pitx2* was exclusively expressed in Ex-4 neurons, but *Vwc2* was expressed in both subsets of neurons (*Figure 2M–N*). These data suggest that the projection-specific SC neurons represent subpopulations of neurons in specific SC layers that can be distinguished by their gene expression profiles.

## Roles of Cbln2+ and Pitx2+ SC neurons in sensory-triggered behaviors related to predator avoidance and prey capture

*Cbln2* and *Pitx2* were highly expressed in LPTN-projecting and ZI-projecting SC neurons, respectively (*Figure 2I*), which also displayed the highest fidelity of Op and InG layer specificity based on our analysis of in situ RNA hybridization images from Allen Brain Atlas (https://mouse.brain-map.org) (*Figure 3—figure supplement 1A*). To examine whether *Cbln2* acts as a key molecular marker of SC circuits associated with predator avoidance, we generated Cbln2-IRES-Cre mouse line (*Figure 3—figure supplement 1B–C*). We first tested whether it is possible to specifically label Cbln2+ SC neurons in Cbln2-IRES-Cre mice by injecting AAV-DIO-EGFP into the SC of these mice. EGFP-expressing neurons were distributed predominantly in the Op layer of the SC (*Figure 3A*). More than 90% of EGFP-expressing SC neurons were positive for *Cbln2* mRNA (91 ± 9%, n=3 mice), and SC neurons expressing *Cbln2* mRNA were predominantly positive for EGFP (92 ± 11%, n=3 mice), suggesting that it is possible to specifically label Cbln2+ neurons in the SC of Cbln2-IRES-Cre mice (*Figure 3B*). In addition, EGFP-expressing SC neurons were mostly positive for *vGlut2* mRNA (93 ± 6%, n=3 mice), and very few were positive for *vGAT* mRNA (4 ± 2.1%, n=3 mice), confirming that the Cbln2+ SC neurons were predominantly glutamatergic (*Figure 3—figure supplement 1D*).

Under laboratory conditions, mice exhibit a freezing response to an overhead moving visual target; this is an innate behavior that may be crucial to the avoidance of aerial predators in the natural environment (*De Franceschi et al., 2016*). We assessed the role of Cbln2+ SC neurons in this behavior (*Figure 3C*) by selectively silencing Cbln2+ SC neurons using tetanus neurotoxin (TeNT) (*Schiavo et al., 1992*). AAV-DIO-EGFP-2A-TeNT was bilaterally injected into the SC of Cbln2-IRES-Cre mice, resulting in the expression of EGFP and TeNT in Cbln2+ SC neurons (*Figure 3—figure supplement 1E*). The effectiveness and specificity of TeNT-mediated synaptic inactivation of SC neurons have been validated in earlier studies (*Shang et al., 2018*; *Shang et al., 2019*). We found that control mice with Cbln2+ SC neurons expressing EGFP (Ctrl) exhibited freezing in response to an overhead moving visual target (*Figure 3—video 1*; black trace in *Figure 3D*). In contrast, synaptic inactivation of Cbln2+ SC neurons by TeNT strongly impaired visually evoked freezing responses (*Figure 3—video 1*; red trace in *Figure 3D*). Quantitative analyses indicated that synaptic inactivation of Cbln2+ SC neurons caused a significant increase in locomotion speed during visual stimuli but not before or after visual stimuli (*Figure 3E–G*). However, inactivation of Cbln2+ SC neurons did not alter the efficiency of predatory hunting (*Figure 3—figure supplement 1F–I*). These data suggest that Cbln2+ SC neurons are selectively required for a visually evoked freezing response in mice.

To examine whether *Pitx2* acts as a key molecular marker of SC circuits for prey capture, we studied the Pitx2-Cre knock-in line (*Liu et al., 2003*). We first examined whether it is possible to specifically label Pitx2+ SC neurons in Pitx2-Cre mice by injecting AAV-DIO-EGFP into the SC of these mice. EGFP-expressing neurons were distributed predominantly in the intermediate layers of the SC (*Figure 3H*). Most EGFP-expressing SC neurons were positive for *Pitx2* mRNA (88 ± 8%, n=3 mice), and SC neurons expressing *Pitx2* mRNA were predominantly positive for EGFP (89 ± 9%, n=3 mice), suggesting that Pitx2+ SC neurons were specifically labeled in Pitx2-Cre mice (*Figure 3I*). Moreover, most EGFP-expressing SC neurons were positive for *vGlut2* mRNA (95 ± 7%, n=3 mice), and very few were positive for *vGAT* mRNA (6 ± 2.5%, n=3 mice), confirming that Pitx2+ SC neurons are predominantly glutamatergic (*Figure 3—figure supplement 1J*).

To explore the role of Pitx2+ SC neurons in prey capture (*Figure 3J*), we injected AAV-DIO-EGFP-2A-TeNT into the SC of Pitx2-Cre mice; this resulted in the expression of EGFP and TeNT in Pitx2+ SC neurons (*Figure 3—figure supplement 1K*). Synaptic inactivation of Pitx2+ SC neurons impaired prey capture (*Figure 3—video 2*; *Figure 3K*) by increasing the latency to attack (*Figure 3L*), prolonging the time required for prey capture (*Figure 3M*), and reducing the frequency of attack (*Figure 3N*). However, the visually evoked freezing response was not impaired in these mice, as evidenced by a lack of significant changes in locomotion speed before, during, or after the

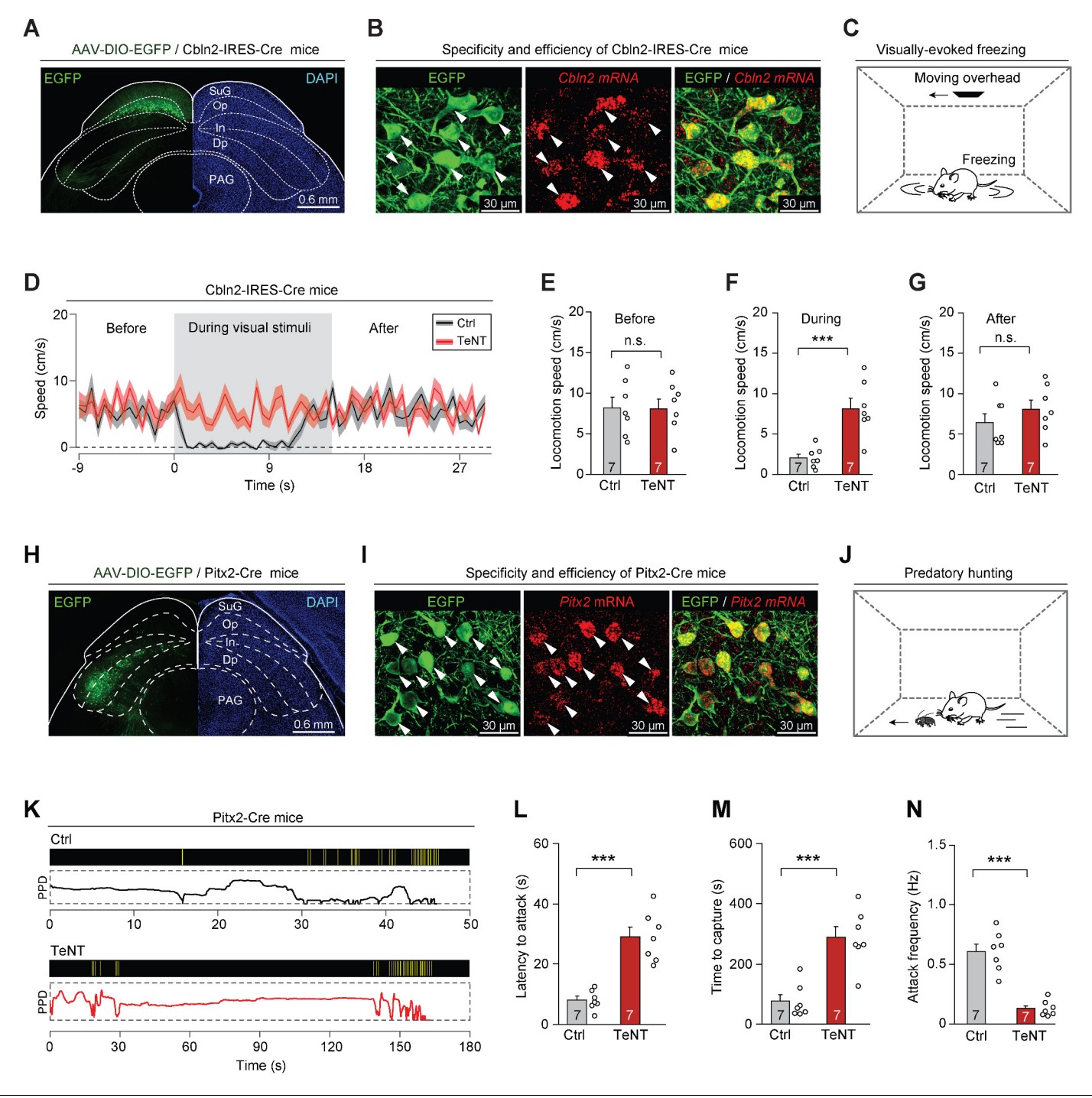

**Figure 3.** Synaptic inactivation of Cbln2+ and Pitx2+ superior colliculus (SC) neurons. (**A**) Sample coronal section showing the restricted distribution of EGFP-expressing neurons in the optic nerve (Op) layer of the SC in Cbln2-IRES-Cre mice. (**B**) Sample micrographs showing the specificity and efficiency of the Cbln2-IRES-Cre line for labeling of SC neurons expressing Cbln2 mRNA. (**C**) Schematic diagram showing the behavioral paradigm of the visually evoked freezing response in mice. (**D**) Time courses of locomotion speed before, during, and after the sweep of an overhead moving visual target in mice without (Ctrl) or with (tetanus neurotoxin [TeNT]) synaptic inactivation of Cbln2+ SC neurons. (**E–G**) Quantitative analysis of locomotion speed before (**E**), during (**F**), and after (**G**) the sweep of an overhead moving target in mice without (Ctrl) and with (TeNT) synaptic inactivation of Cbln2+ SC neurons. (**H**) Sample coronal section showing the restricted distribution of EGFP-expressing neurons in the In layer of the SC in Pitx2-Cre mice. (**I**) Sample micrographs showing the specificity and efficiency of the Pitx2-Cre line for labeling of SC neurons expressing Pitx2 mRNA. (**J**) Schematic diagram showing the behavioral paradigm of predatory hunting in mice. (**K**) Behavioral ethograms of predatory hunting in mice without (Ctrl) and with (TeNT) synaptic inactivation of Pitx2+ SC neurons. The yellow vertical lines indicate jaw attacks. The PPD curve shows the time course of prey-predator

*Figure 3 continued on next page*

*Figure 3 continued*

distance. (**L–N**) Quantitative analysis of latency to attack (**L**), time to capture (**M**), and attack frequency (**N**) in mice without (Ctrl) and with (TeNT) synaptic inactivation of SC Pitx2+ neurons. The data in (**D–G, L–N**) are presented as mean ± SEM (error bars). The statistical analyses in (**E–G, L–N**) were performed using Student's t-test (n.s. p>0.1; ***p < 0.001). For the p-values, see *Supplementary file 8*. Scale bars are indicated in the graphs.

The online version of this article includes the following video and figure supplement(s) for figure 3:

**Figure supplement 1.** Generation of Cbln2-IRES-Cre mice to test the function of Cbln2+ and Pitx2+ superior colliculus (SC) neurons.

**Figure 3—video 1.** An example video showing that synaptic inactivation of Cbln2+ SC neurons by TeNT impaired visually-evoked freezing responses.

https://elifesciences.org/articles/69825#fig3video1

**Figure 3—video 2.** An example video showing that synaptic inactivation of Pitx2+ SC neurons by TeNT impaired prey capture behavior in the arena.

https://elifesciences.org/articles/69825#fig3video2

presentation of visual stimuli (*Figure 3—figure supplement 1L–O*). These data suggest that Pitx2+ SC neurons are selectively required for prey capture behavior in mice.

## Cbln2+ and Pitx2+ SC neurons encode distinct sensory stimuli

Next, we addressed how Cbln2+ and Pitx2+ SC neurons participate in predator avoidance and prey capture. Rodents use vision to detect aerial predators (*De Franceschi et al., 2016*; *Yilmaz and Meister, 2013*), whereas they use vibrissal tactile information (*Anjum et al., 2006*) and vision (*Hoy et al., 2016*) for prey capture. To examine whether Cbln2+ and Pitx2+ SC neurons process visual and vibrissal tactile information, we expressed GCaMP7 in these neurons and implanted an optical fiber above the neurons (*Figure 4A–B*; *Dana et al., 2019*; *Gunaydin et al., 2014*). Please note that in this study, only Cbln2+ neurons in medial SC were recorded because medial SC preferentially monitor upper visual field to detect aerial predator (*Ito and Feldheim, 2018*). We provided visual and vibrissal tactile stimuli to head-fixed mice standing on a treadmill and simultaneously performed fiber photometry to record GCaMP fluorescence in these neurons (*Figure 4C*). The visual stimulus was a computer-generated black circle (5° or 25° in diameter) moving at a controlled velocity (32°/s or 128°/s) across the visual receptive field (RF) on a tangent screen (*Shang et al., 2019*). The vibrissal tactile stimuli, which were designed to mimic the tactile cues produced by moving prey, were brief gentle air puffs (100 ms, 0–40 p.s.i.) directed toward the vibrissal region contralateral or ipsilateral to the recorded side (*Shang et al., 2019*).

We found that the centers of the visual RFs of Cbln2+ SC neurons were distributed predominantly in the dorsal quadrants of the visual field (*Figure 4—figure supplement 1A*). These neurons responded broadly to visual stimuli moving in various directions, with a preference for the temporal-to-nasal direction (*Figure 4D,H*). In addition, they responded more strongly to circles moving at lower velocity (*Figure 4E,I*) and those with smaller diameters (*Figure 4F,J*). However, the Cbln2+ SC neurons did not respond to air puffs applied to the vibrissal area (*Figure 4—figure supplement 1B*). Unlike Cbln2+ SC neurons, Pitx2+ SC neurons responded to air puffs directed toward the vibrissal region contralateral to the recorded side (*Figure 4G,K*). However, they did not respond to moving visual stimuli (*Figure 4—figure supplement 1C–E*). These data indicate that Cbln2+ SC neurons may specifically process visual information derived from an aerial cruising predator, while Pitx2+ SC neurons may be selectively involved in processing vibrissal tactile stimuli mimicking moving prey. Thus, Cbln2+ and Pitx2+ SC neurons may comprise two distinct sets of circuit modules that are used to detect visual cues produced by aerial predator and vibrissal cues produced by terrestrial prey of mice.

## Cbln2+ and Pitx2+ SC neurons receive distinct monosynaptic inputs

Next, we performed monosynaptic retrograde tracing using recombinant rabies virus (RV) (*Wickersham et al., 2007*) to examine how Cbln2+ and Pitx2+ SC neurons are connected with neural structures associated with sensory information processing (*Figure 5A–C*, *Figure 5—figure supplement 1A–B*). A brain-wide survey revealed a number of monosynaptic projections to Cbln2+ and Pitx2+ SC neurons (*Figure 5D–G*, *Figure 5—figure supplement 1C–F*). First, Cbln2+ SC neurons were monosynaptically innervated by a subset of retinal ganglion cells in the contralateral retina (*Figure 5D*) and neurons in layer 5 of the ipsilateral primary visual cortex (V1) (*Figure 5E*). In contrast, Pitx2+ SC neurons did not receive monosynaptic inputs from these visual structures

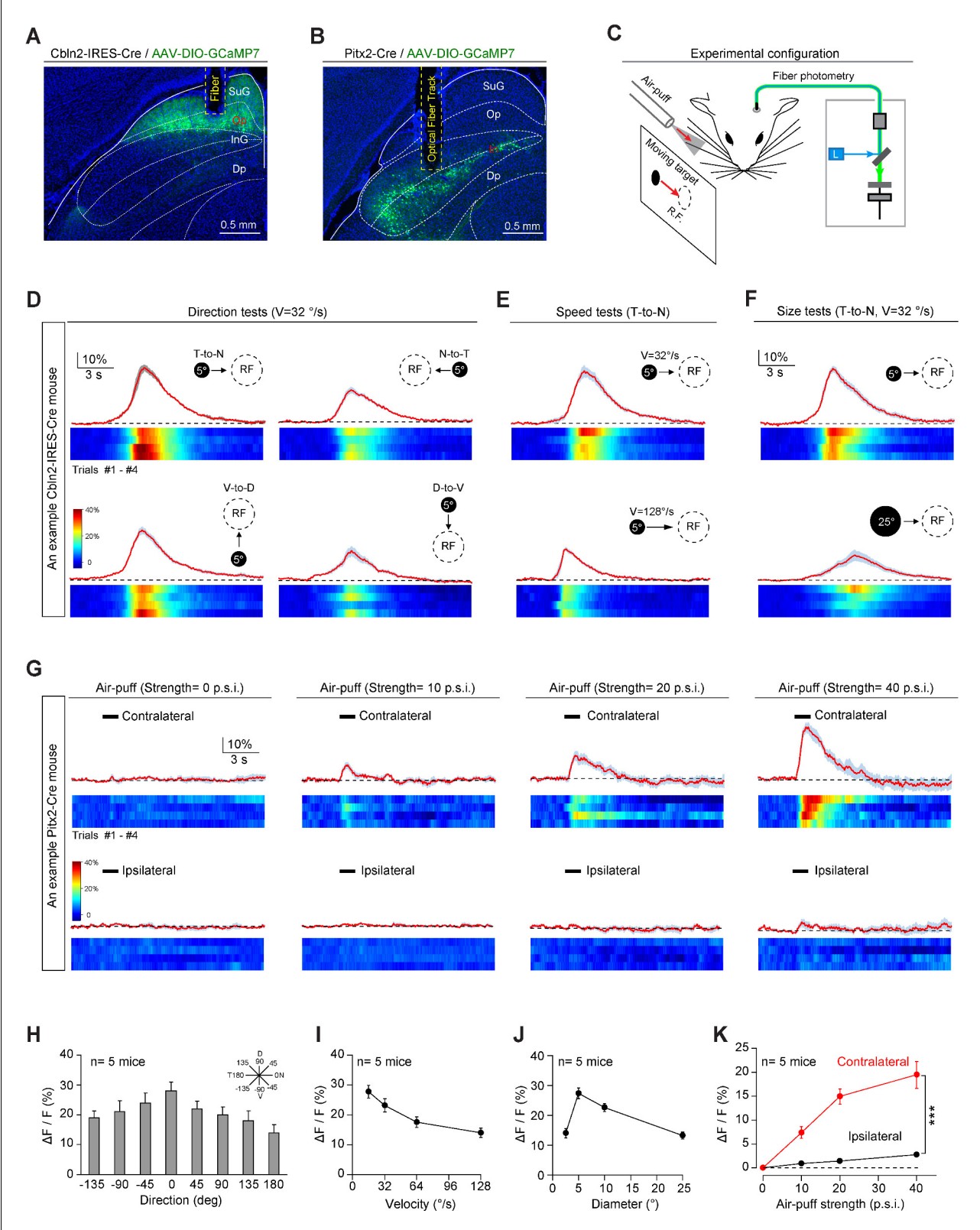

**Figure 4.** Sensory responses of Cbln2+ and Pitx2+ superior colliculus (SC) neurons. (**A, B**) Sample micrographs showing the optical fiber tracks above GCaMP7-positive SC neurons in Cbln2-IRES-Cre (**A**) and Pitx2-Cre (**B**) mice. (**C**) Schematic diagram of the experimental configuration showing vibrissal tactile stimulation (air puff) and visual stimulation; the latter was presented as a black circle moving across the receptive field (RF) on a tangent screen. (**D**) Normalized GCaMP fluorescence changes (ΔF/F) and heatmaps of Cbln2+ SC neurons in an example mouse in response to a stimulus consisting of

*Figure 4 continued on next page*

Figure 4 continued

a black circle (5°) moving at 32°/s in various directions (T-to-N, N-to-T, V-to-D, D-to-V). N, D, T, and V indicate nasal, dorsal, temporal, and ventral, respectively. (E) Normalized GCaMP fluorescence changes (ΔF/F) and heatmaps of Cbln2+ SC neurons in an example mouse in response to a black circle (5°) moving (T-to-N) at different velocities (32°/s and 128°/s). (F) Normalized GCaMP fluorescence changes (ΔF/F) and heatmaps of Cbln2+ SC neurons in an example mouse in response to black circles of different sizes (5° and 25°) moving in a T-to-N direction at 32°/s. (G) Normalized GCaMP fluorescence changes (ΔF/F) and heatmaps of Pitx2+ SC neurons in an example mouse in response to air puffs of different strengths (0, 10, 20, and, 40 p.s.i.) directed toward the contralateral or ipsilateral vibrissal area. (H) Quantitative analysis of peak GCaMP responses of Cbln2+ SC neurons to black circles moving in eight directions. Inset, eight directions spaced by 45°. (I) Quantitative analysis of the peak GCaMP responses of Cbln2+ SC neurons to black circles moving at different velocities. (J) Quantitative analysis of the peak GCaMP responses of Cbln2+ SC neurons to moving black circles with different diameters. (K) Quantitative analysis of the peak GCaMP responses of Pitx2+ SC neurons to air puffs of different strengths directed toward the contralateral or ipsilateral vibrissal areas. The data in (D–K) are presented as mean ± SEM (error bars). The statistical analyses in (K) were performed by one-way ANOVA (***p < 0.001). For the p-values, see *Supplementary file 8*. Scale bars are indicated in the graphs.

The online version of this article includes the following figure supplement(s) for figure 4:

**Figure supplement 1.** Sensory response properties of Cbln2+ and Pitx2+ superior colliculus (SC) neurons.

(*Figure 5D–E*). Second, Pitx2+ SC neurons, but not Cbln2+ SC neurons, received robust monosynaptic inputs from the subnuclei of the trigeminal complex (Pr5 and Sp5) (*Figure 5F*, *Figure 5—figure supplement 1C*), the primary somatosensory cortex (S1) (*Figure 5G*), and the ZI (*Figure 5—figure supplement 1D*), which are involved in processing tactile information. Third, Pitx2+ SC neurons, but not Cbln2+ SC neurons, also received monosynaptic inputs from motor-related brain areas (e.g., SNr and M1/M2) (*Figure 5—figure supplement 1E–F*) and the cingulate cortex (Cg1/2) (*Figure 5—figure supplement 1F*). Quantitative analysis of retrogradely labeled cells in various brain areas indicated that Cbln2+ and Pitx2+ SC neurons may receive two sets of presynaptic inputs that are mutually exclusive (*Figure 5H*). These morphological data support the hypothesis that Cbln2+ and Pitx2+ SC neurons belong to two distinct sets of circuit modules that are used to detect visual cues produced by aerial predators (retina and V1) and tactile cues produced by terrestrial prey (Pr5, Sp5, S1, and ZI) of mice, respectively.

## Target-specific projections of Cbln2+ and Pitx2+ SC neurons

To compare the brain-wide projections of Cbln2+ and Pitx2+ SC neurons, we injected AAV-DIO-EGFP into Cbln2-IRES-Cre and Pitx2-Cre mice (*Figure 6A*), resulting in the expression of EGFP in Cbln2+ and Pitx2+ neurons that were distributed in distinct laminae of the SC (*Figure 6B*). In addition to the LPTN and the ZI, Cbln2+ and Pitx2+ SC neurons sent divergent projections to a number of areas in the thalamus, midbrain, pons, and medulla (*Figure 6C–F*). Strikingly, their downstream target brain areas rarely overlapped (*Figure 6C–F*), supporting the hypothesis that these two subtypes of SC neurons are functionally distinct.

Then, we focused on characterizing their projections to the LPTN and ZI. In the LPTN, we found strong EGFP+ axonal projections from Cbln2+ SC neurons but not from Pitx2+ SC neurons (*Figure 7A*). Quantitative analyses of EGFP fluorescence indicated that the density of EGFP+ axons from Cbln2+ SC neurons was significantly higher than that from Pitx2+ SC neurons in the LPTN (*Figure 7B*). These data suggest that Cbln2+ SC neurons, but not Pitx2+ SC neurons, send strong projections to the LPTN. Similarly, in the ZI, we found strong EGFP+ axonal projections from Pitx2+ SC neurons but not from Cbln2+ SC neurons (*Figure 7C*). Quantitative analysis of the EGFP fluorescence indicated that, in the ZI, the density of EGFP+ axons from Pitx2+ SC neurons was significantly higher than that of EGFP+ axons from Cbln2+ SC neurons (*Figure 7D*). These data suggest that Pitx2+ SC neurons, but not Cbln2+ SC neurons, send strong projections to the ZI.

We also explored the target-specific projections of Cbln2+ and Pitx2+ SC neurons using retrograde AAV (*Tervo et al., 2016*). AAV2-retro-DIO-EGFP was injected into the LPTN of Cbln2-IRES-Cre mice, followed by injection of AAV-DIO-mCherry into the SC (*Figure 7E*). More than 80% of mCherry+ SC neurons (83 ± 9%, n=3 mice) were labeled by EGFP, suggesting that a large proportion of Cbln2+ SC neurons project to the LPTN (*Figure 7F–G*). We injected AAV2-retro-DIO-EGFP into the ZI of Pitx2-Cre mice, followed by injection of AAV-DIO-mCherry into the SC (*Figure 7H*). More than two-thirds of mCherry+ SC neurons (72 ± 11%, n=3 mice) were labeled by EGFP, suggesting that a large proportion of Pitx2+ neurons project to the ZI (*Figure 7I–J*).

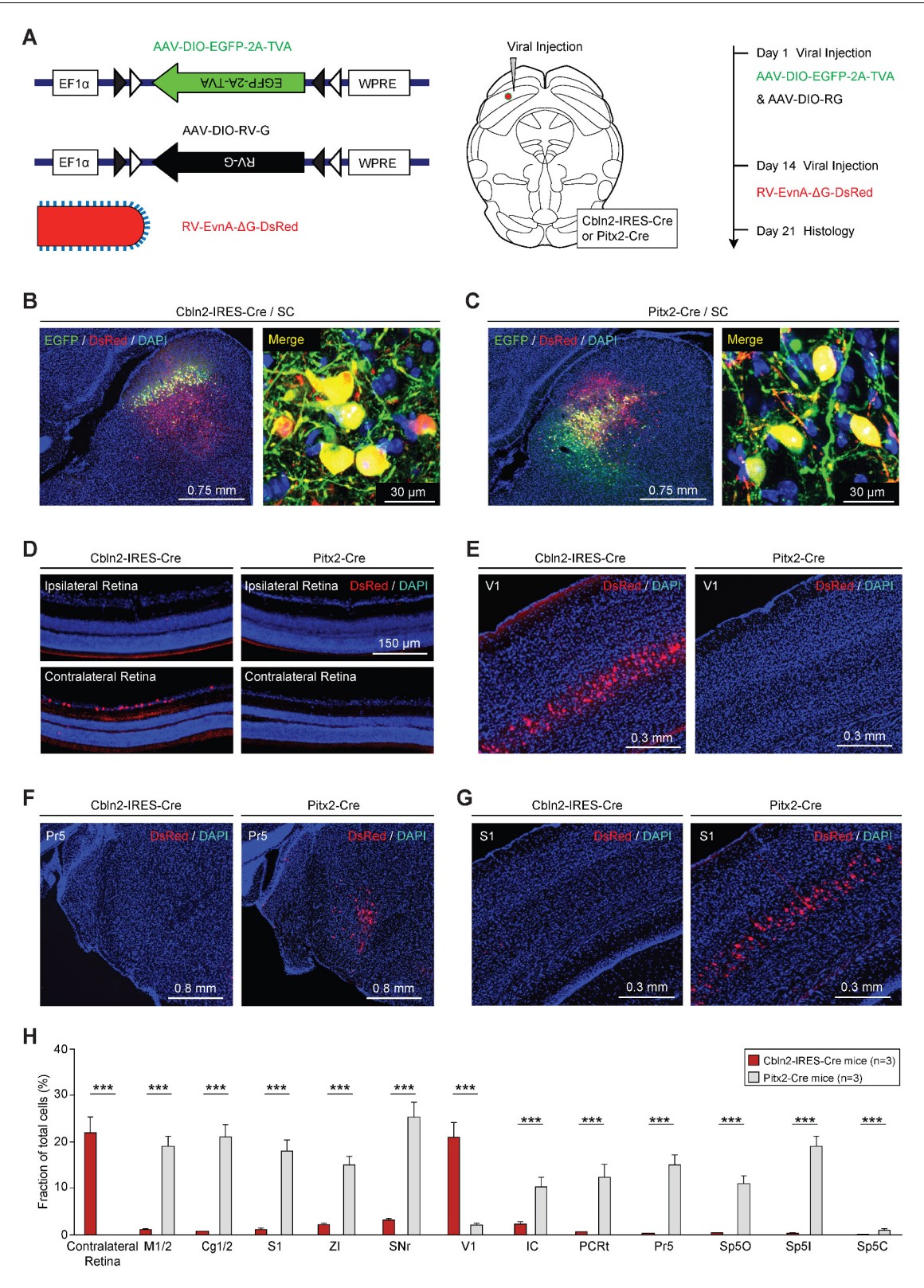

**Figure 5.** Retrograde tracing of Cbln2+ and Pitx2+ superior colliculus (SC) neurons using rabies virus (RV). (**A**) Series of schematic diagrams showing the strategy for monosynaptic retrograde tracing of Cbln2+ and Pitx2+ SC neurons using a combination of AAV and RV. Left, AAV helpers and RV used for injection. Middle, injection into the SC of Cbln2-IRES-Cre and Pitx2-Cre mice. Right, timing of AAV and RV injections. (**B, C**) Sample micrographs showing the expression of EGFP (green) and DsRed (red) in Cbln2+ and Pitx2+ neurons in the SC of Cbln2-IRES-Cre (**B**) and Pitx2-Cre mice (**C**). The

*Figure 5 continued on next page*

*Figure 5 continued*

dually labeled cells indicate starter cells. For single-channel images, see *Figure 5—figure supplement 1*. (D–G) Sample micrographs showing DsRed+ cells in various brain regions, including the contralateral and ipsilateral retina (D), the primary visual cortex (V1) (E), the contralateral principal trigeminal nucleus (Pr5) (F), and the ipsilateral primary somatosensory cortex (S1) (G), of Cbln2-IRES-Cre and Pitx2-Cre mice. (H) Fractional distribution of total DsRed-labeled cells in various brain regions that monosynaptically project to Cbln2+ and Pitx2+ SC neurons. The scale bars are labeled in the graphs. The number of mice (H) is indicated in each graph. The data in (H) are presented as mean ± SEM. The statistical analyses in (H) were performed using Student's t-test (***p < 0.001). For the p-values, see *Supplementary file 8*.

The online version of this article includes the following figure supplement(s) for figure 5:

**Figure supplement 1.** Rabies virus (RV) tracing of Cbln2+ and Pitx2+ superior colliculus (SC) neurons.

## Activation of the Cbln2+ SC-LPTN pathway and the Pitx2+ SC-ZI pathway

The above data indicate that Cbln2+ and Pitx2+ neurons in the SC selectively project to the LPTN and to the ZI, respectively. To test the behavioral relevance of the Cbln2+ SC-LPTN pathway, we injected AAV-DIO-ChR2-mCherry into the SC of Cbln2-IRES-Cre mice (*Boyden et al., 2005*), followed by implantation of an optical fiber above the LPTN (*Figure 8A*). In acute SC slices, light stimulation (10 or 20 Hz, 2 ms) effectively evoked action potential firing in neurons expressing ChR2-mCherry (*Figure 8—figure supplement 1A–B*). We found that light stimulation of ChR2-mCherry+ axon terminals of Cbln2+ SC neurons in the LPTN induced a freezing response in mice (*Figure 8—video 1*, *Figure 8B–C*), as evidenced by a selective reduction of locomotion speed during light stimulation (*Figure 8D*). However, activation of this pathway did not alter the efficiency of prey capture during predatory hunting (*Figure 8—figure supplement 1D–E*). These data indicate that activation of the Cbln2+ SC-LPTN pathway selectively triggers predator avoidance rather than prey capture.

Next, we examined the behavioral relevance of the Pitx2+ SC-ZI pathway. AAV-DIO-ChR2-mCherry was injected into the SC of Pitx2-Cre mice, followed by implantation of an optical fiber above the ZI (*Figure 8E*). In acute SC slices, light stimulation (10 or 20 Hz, 2 ms) evoked phase-locked action potential firing by neurons expressing ChR2-mCherry (*Figure 8—figure supplement 1C*). Photostimulation of ChR2-mCherry+ axon terminals of Pitx2+ SC neurons in the ZI did not evoke a freezing response (*Figure 8—figure supplement 1F–G*). However, activation of this pathway promoted predatory hunting (*Figure 8—video 2*; *Figure 8F–G*) by decreasing the latency to hunt (*Figure 8H*), reducing the time required for prey capture (*Figure 8I*), and increasing the frequency of predatory attacks (*Figure 8J*). These data indicate that activation of the Pitx2+ SC-ZI pathway selectively promotes prey capture without inducing predator avoidance.

## Discussion

The SC of the midbrain is a classical model for the study of early sensorimotor transformation related to sensory-triggered innate behaviors (fish [*Bianco and Engert, 2015*]; rodents [*Dean et al., 1989*]; primates [*Sparks, 1986*]). In mice, a series of projection-defined SC circuits have been linked to sensory-triggered innate behaviors such as predator avoidance (*Evans et al., 2018*; *Shang et al., 2018*; *Shang et al., 2015*; *Wei et al., 2015*; *Zhou et al., 2019*) and prey capture (*Hoy et al., 2019*; *Shang et al., 2019*). However, the number of cell types in the SC, their molecular signatures, their projection patterns, and their functional roles in these behaviors remain unclear. Here, we used a combined approach consisting of single-cell transcriptomic analysis and circuit analysis to address the above questions.

Using high-throughput single-cell transcriptomic analyses, we first systematically analyzed the cell-type diversity of the SC. We found that the majority of neuronal subtypes defined by gene expression patterns showed layer-specific distribution in the SC (*Figure 1*). To better understand the correlation between projection patterns and transcriptomics, we performed projection-specific scRNA-seq of cells in the SC-LPTN and SC-ZI pathways and identified Cbln2 and Pitx2 as key molecular markers for SC neurons with different projections (*Figure 2*). Cbln2+ and Pitx2+ SC neurons, distributed in distinct layers of the SC, were functionally involved in predator avoidance and prey capture behavior, respectively (*Figure 3*). Strikingly, these two neuronal subtypes processed neuroethological information received through distinct sensory modalities (*Figure 4*) and were connected

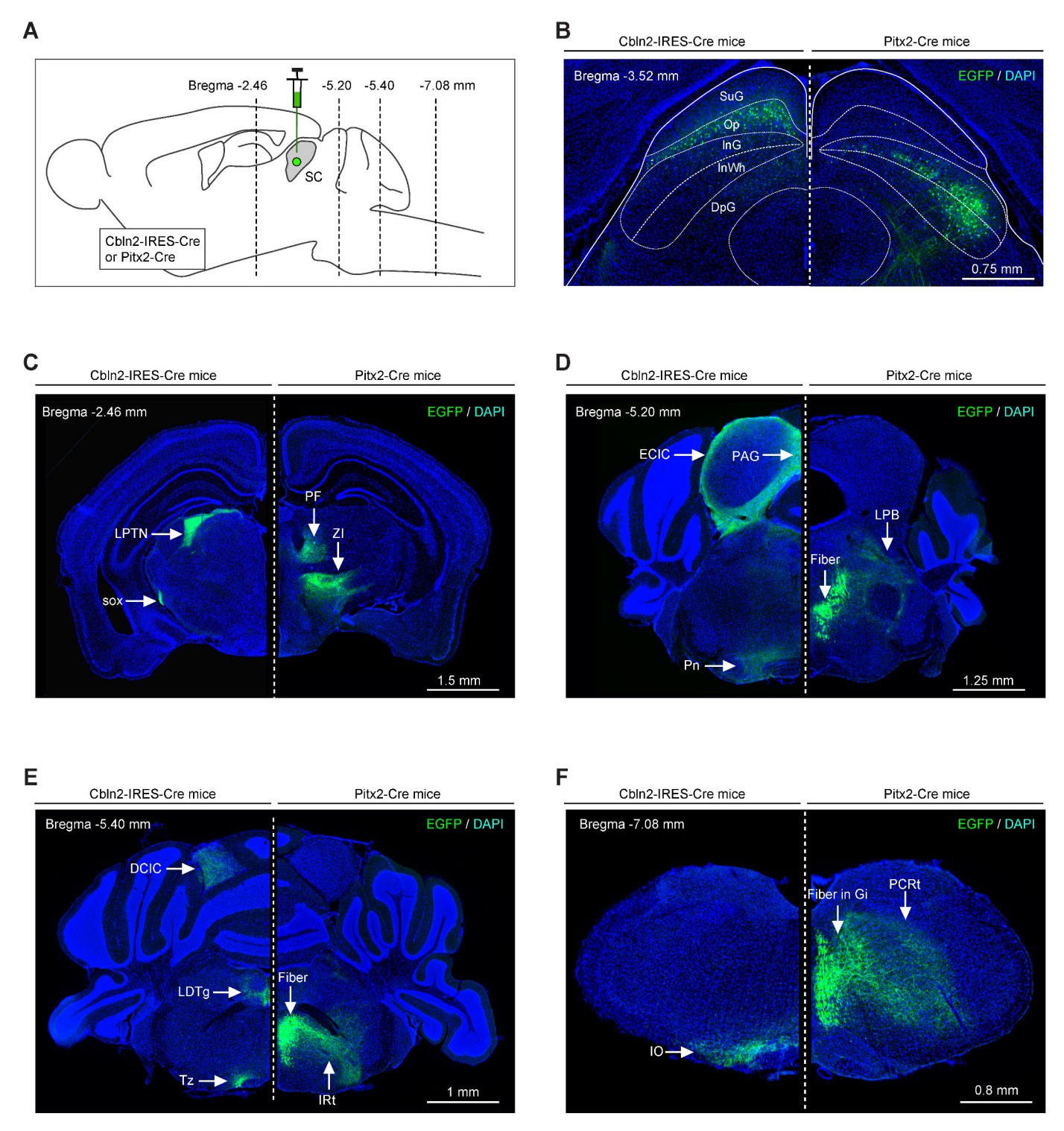

**Figure 6.** Efferent projections of Cbln2+ and Pitx2+ superior colliculus (SC) neurons are segregated. (A) Schematic diagram showing the strategy to map the efferent projections of Cbln2+ and Pitx2+ SC neurons. (B) Example coronal sections of Cbln2-IRES-Cre and Pitx2-Cre mice showing the distribution of infected neurons in the SC. (C–F) Example micrographs showing EGFP+ axons of Cbln2+ and Pitx2+ SC neurons in the target brain regions at the level of thalamus (C), midbrain (D), pons (E), and medulla (F). Abbreviations: LPTN, lateral posterior thalamic nucleus; PF, parafascicular nucleus; ZI, zona incerta; Pn, pontine nucleus; LPB, lateral parabrachial nucleus; IO, inferior olive; PCRt, parvicellular reticular nucleus; LDTg, laterodorsal tegmental nucleus; IRt, intermediate reticular nucleus; PAG, periaqueductal gray; ECIC, external cortex of the inferior colliculus; Tz, nucleus of the trapezoid body. Scale bars are labeled in the graphs.

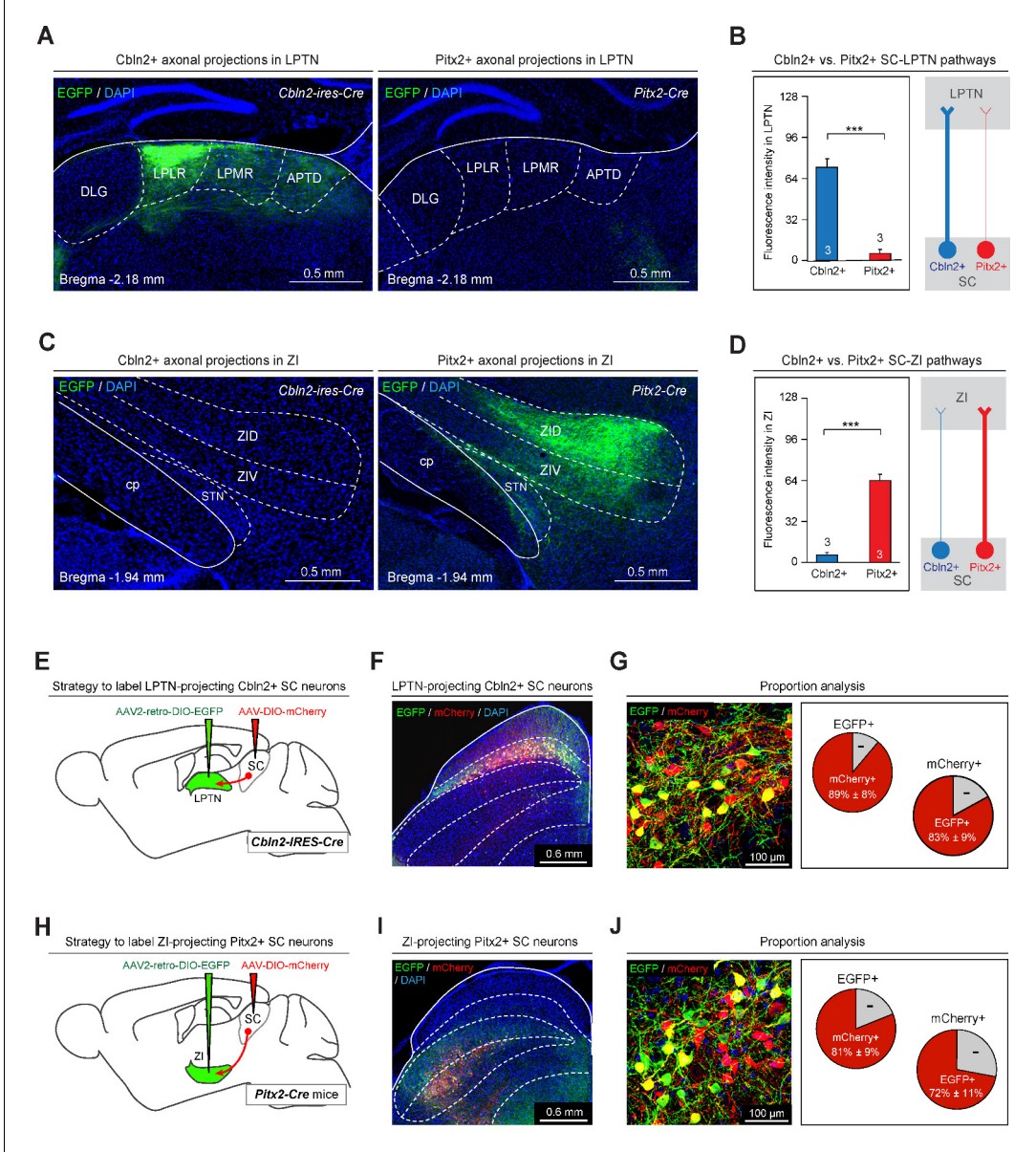

**Figure 7.** Anterograde and retrograde mapping of Cbln2+ superior colliculus (SC)-lateral posterior thalamic nucleus (LPTN) and Pitx2+ SC-zona incerta (ZI) pathways. (**A**) Sample micrographs showing the distribution of EGFP-positive axons in the LPLR and LPMR (collectively the LPTN) of Cbln2-IRES-Cre (left) and Pitx2-Cre (right) mice. (**B**) Quantitative analysis of fluorescence signals from EGFP+ axons in the LPTN of Cbln2-IRES-Cre and Pitx2-Cre mice. (**C**) Sample micrographs showing the distribution of EGFP-positive axons in the ZI of Cbln2-IRES-Cre (left) and Pitx2-IRES-Cre (right) mice. (**D**) Quantitative analysis of fluorescence signals from EGFP+ axons in the ZI of Cbln2-IRES-Cre and Pitx2-Cre mice. (**E**) Schematic diagram showing the viral injection strategy used to label LPTN-projecting Cbln2+ SC neurons. (**F**) Coronal section from a Cbln2-IRES-Cre mouse showing the distribution of Cbln2+ SC neurons labeled by AAV2-retro-DIO-EGFP and AAV-DIO-mCherry. (**G**) Sample micrograph (left) and quantitative analysis (right) showing the number of LPTN-projecting Cbln2+ SC neurons (EGFP+) relative to total Cbln2+ SC neurons (mCherry+). (**H**) Schematic diagram showing the viral injection strategy used to label ZI-projecting Pitx2+ SC neurons. (**I**) Coronal section from a Pitx2-Cre mouse showing the distribution of Pitx2+ SC neurons labeled by AAV2-retro-DIO-EGFP and AAV-DIO-mCherry. (**J**) Sample micrograph (left) and quantitative analysis (right) showing the number of ZI-projecting Pitx2+ SC neurons (EGFP+) relative to total Pitx2+ SC neurons (mCherry+). The data in A, B, D, F, I, and L are presented as mean ± SEM (error bars). The statistical analyses in D and F were performed using Student's t-test (***p < 0.001). For the p-values, see *Supplementary file 8*. Scale bars are indicated in the graphs.

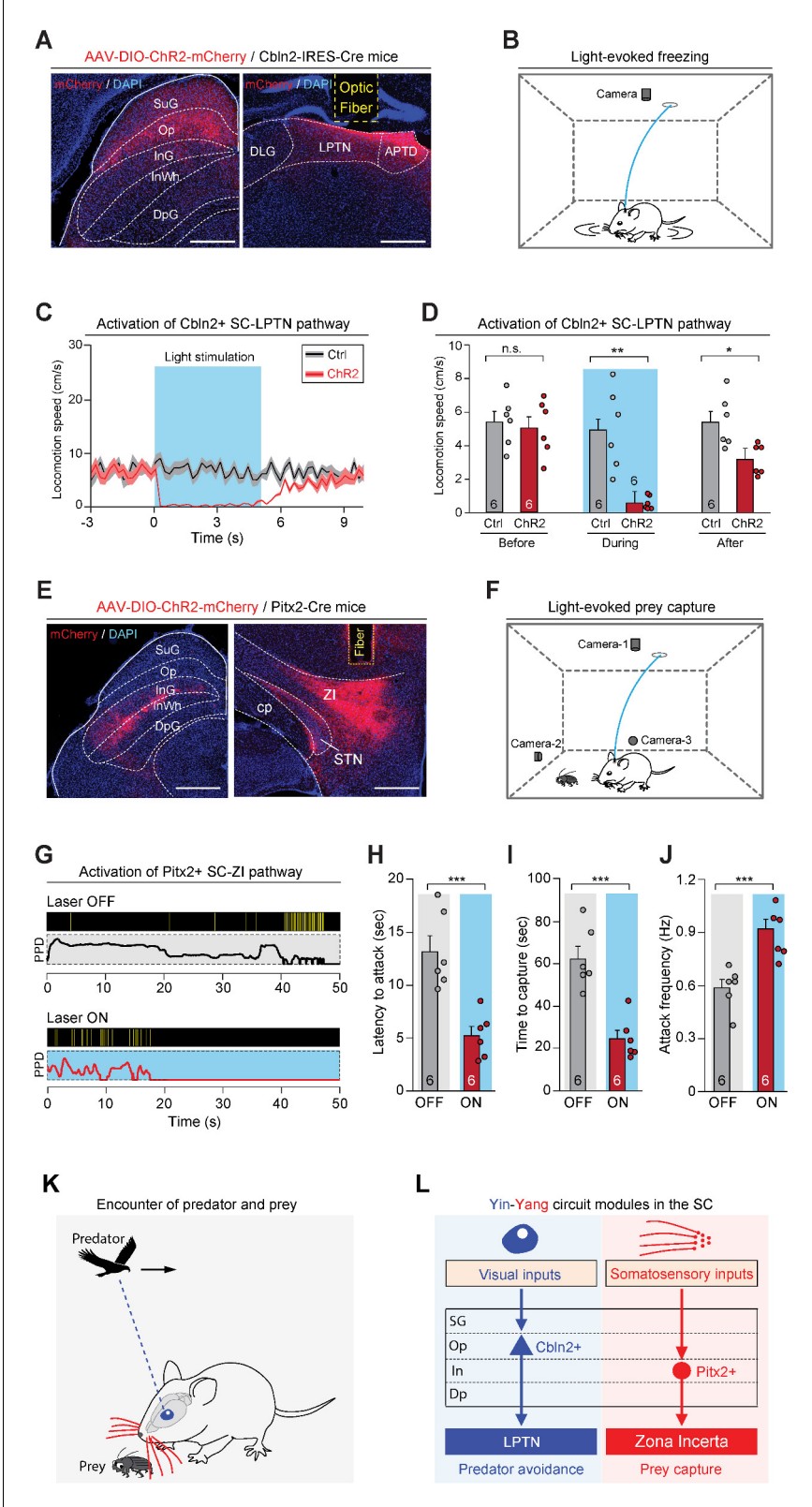

**Figure 8.** Activation of the Cbln2+ superior colliculus (SC)-lateral posterior thalamic nucleus (LPTN) and Pitx2+ SC-zona incerta (ZI) pathways. (**A**) Sample micrographs showing the expression of ChR2-mCherry in the Cbln2+ SC neurons of Cbln2-IRES-Cre mice (left) and the optical fiber track above the ChR2-mCherry+ axons in the LPTN (right). (**B**) Schematic diagram showing the behavioral paradigm for the light-evoked freezing response in an arena.

*Figure 8 continued on next page*

*Figure 8 continued*

(C) Time courses of the locomotion speed of mice before, during, and after light stimulation (10 Hz, 20 ms, 5 mW, 5 s) of the Cbln2+ SC-LPTN pathway expressing ChR2-mCherry (ChR2) or mCherry (Ctrl). (D) Quantitative analysis of the locomotion speed of mice before, during, and after light stimulation of the Cbln2+ SC-LPTN pathway expressing ChR2-mCherry (ChR2) or mCherry (Ctrl). (E) Sample micrographs showing the expression of ChR2-mCherry in the Pitx2+ SC neurons of Pitx2-Cre mice (left) and the optical fiber track above the ChR2-mCherry+ axons in the ZI (right). (F) Schematic diagram showing the behavioral paradigm for prey capture paired with light stimulation of the Pitx2+ SC-ZI pathway. (G) Behavioral ethograms of predatory hunting in mice without (laser OFF) and with (laser ON) light stimulation of the Pitx2+ SC-ZI pathway. (H–J) Quantitative analyses of latency to attack (H), time to capture (I), and attack frequency (J) in mice without (OFF) and with (ON) light stimulation of the Pitx2+ SC-ZI pathway. (K) Schematic diagram showing a mouse encountering a cruising aerial predator or a terrestrial prey in the natural environment. (L) Yin-Yang circuit modules formed by Cbln2+ and Pitx2+ SC neurons and their downstream target areas. The Cbln2+ SC neurons in the 'Yin' module detect the sensory features of cruising aerial predators and initiate freezing as a defensive response for the avoidance of predators through the Cbln2+ SC-LPTN pathway. The Pitx2+ SC neurons in the Yang module mediate tactile-triggered prey capture behavior through the Pitx2+ SC-ZI pathway. The data in (C, D, H–J) are presented as mean ± SEM (error bars). The statistical analyses in (D, H–J) were performed using Student's t-test (n.s. p>0.1; *p<0.05; **p < 0.01; ***p<0.001). For the p-values, see *Supplementary file 8*. Scale bars are indicated in the graphs.
The online version of this article includes the following video and figure supplement(s) for figure 8:

**Figure supplement 1.** Activation of Cbln2+ superior colliculus (SC)-lateral posterior thalamic nucleus (LPTN) pathway and Pitx2+ SC-zona incerta (ZI) pathway.
**Figure 8—video 1.** An example video showing that light stimulation of ChR2-mCherry+ axon terminals of Cbln2+ SC neurons in the LPTN induced freezing response in mice.
https://elifesciences.org/articles/69825#fig8video1
**Figure 8—video 2.** An example video showing light stimulation of ChR2-mCherry+ axon terminals of Pitx2+ SC neurons in the ZI promoted predatory hunting behavior in mice, related to *Figure 8F–G*.
https://elifesciences.org/articles/69825#fig8video2

to almost completely different sets of input-output synaptic connectomes (*Figures 5*, *6* and *7*). Finally, activation of the Cbln2+ SC-LPTN and the Pitx2+ SC-ZI pathways copied the behavioral phenotypes of SC-LPTN and SC-ZI pathways (*Figure 8*).

We believe our data allow two conclusions. First, they reveal that it is the transcriptomically defined neuronal subtypes and their projections, when combined together, define early sensorimotor transformation and the subsequent behavior. This finding supports the hypothesis that transcriptomically defined circuit modules correspond to specific behaviors. Previous studies of this 'correspondence' question in the hypothalamus and prefrontal cortex, two brain areas that are involved in the regulation of motivation and cognition, did not find a clear correspondence between transcriptomically defined neurons and behaviors (*Kim et al., 2019*; *Lui et al., 2021*; *Moffitt et al., 2018*). In combination with these pioneering studies, our results suggest that the circuit design of sensorimotor transformation in the midbrain, which requires precise detection of sensory features and rapid initiation of innate behaviors, may differ from the circuit design in brain areas that are responsible for complex information processing such as that related to the regulation of motivation and cognition.

Second, our data identified Cbln2 and Pitx2 as discrete markers that label neurons in the Op and InG layers of the SC. Intriguingly, we found three Op subtypes (two subtypes of excitatory neurons; one subtype of inhibitory neurons) and three InG/InWh subtypes (two subtypes of excitatory neurons; one subtype of inhibitory neurons) in our unsupervised high-throughput single-cell transcriptome and spatial information analyses. Cbln2 was extensively expressed by two subtypes of Op layer excitatory neurons, but Pitx2 was expressed by only one subtype of InG layer excitatory neurons, indicating the heterogeneity of neurons located in different SC layers. Our data also suggest that neurons with similar transcriptomic profiles may tend to participate in the same projection paths and to play a role in the regulation of behavior.

Cbln2 and other members of the Cerebellin family are synaptic organizer molecules that bind to presynaptic neurexins and to postsynaptic receptors (*Cheng et al., 2016*). Recent studies have shown that Cbln2 may participate in synapse formation (*Matsuda and Yuzaki, 2011*; *Seigneur and Südhof, 2018*). It remains to be determined whether Cbln2 in Cbln2-expressing SC neurons

participates in the formation of the SC-LPTN pathway. As a transcription factor, Pitx2 participates in the migration of collicular neurons during brain development (*Waite et al., 2013*). In the adult brain, activation of Pitx2+ SC neurons elicited stereotyped head displacements in a body-referenced frame (*Masullo et al., 2019*). However, the functional role of Pitx2+ SC neurons in naturalistic behavioral context has not been demonstrated in this study. Our study has shown that Pitx2+ SC neurons specifically participate in predatory hunting, a goal-directed behavior that occurs in natural environment.

An interesting finding of our study is that the brain-wide projectomes of Cbln2+ and Pitx2+ SC neurons are spatially segregated. Although we only examined the functions of Cbln2+ and Pitx2+ SC-LPTN and SC-ZI pathways, we do not rule out the involvement of other pathways in predator avoidance and prey capture behaviors. In addition to the LPTN, the Cbln2+ SC neurons also project to the inferior colliculus (IC), laterodorsal tegmental nucleus (LDTg), and the inferior olive (IO). It is likely that the Cbln2+ SC-IC pathway may be involved in visual-auditory integration during innate fear in complex environment with both visual and auditory stimuli. The Cbln2+ SC-LDTg pathway may participate in the integration of innate fear responses evoked by olfaction and vision, because the interneurons in the LDTg are involved in olfactory-evoked innate fear (*Yang et al., 2016*). In addition to the ZI, the Pitx2+ SC neurons project to the parafascicular nucleus (PF), intermediate reticular nucleus (IRt), and parvicellular reticular nucleus (PCRt). The Pitx2+ SC-PF pathway may be involved in transmitting prey-relevant tactile signals to the dorsal striatum, a major component of the basal ganglia, for coordination of body movements, because the PF is known as a major input to the dorsal striatum (*Mandelbaum et al., 2019*). The Pitx2+ SC-PCRt pathway may be involved in biting actions during predatory attack, as PCRt contains a large population of craniofacial premotor neurons (*Stanek et al., 2014*).

Our results also raise new questions. First, it is unclear how the genes expressed by Cbln2+ and Pitx2+ SC neurons participate in the formation and function of the Cbln2+ SC-LPTN and Pitx2+ SC-ZI pathways. Single-cell transcriptomic analyses of Cbln2+ and Pitx2+ SC neurons at different developmental stages and genetic manipulations of these neurons may be performed to address this question. Second, the neural substrate that mediates the interactions between the two distinct circuit modules for predator avoidance and prey capture remains to be studied. With the identification of Cbln2+ and Pitx2+ SC neurons, their distinct input-output synaptic connectomes, and the demonstration of their roles in sensory-triggered innate behaviors in hand, these questions can now be addressed.

# Materials and methods

## Key resources table

| Reagent type (species) or resource | Designation | Source or reference | Identifiers | Additional information |
|---|---|---|---|---|
| Strain, strain background (*Mus musculus*) | *Pitx2-Cre* mice | Mutant Mouse Resource Center | Cat# 000126-UCD, RRID:MMRRC_000126-UCD | |
| Strain, strain background (*Mus musculus*) | *vGlut2-IRES-Cre* mice | JAX Mice | Cat# JAX:028863, RRID:IMSR_JAX:028863 | |
| Strain, strain background (*Mus musculus*) | *Cbln2-IRES-Cre* mice | NIBS | NA | |
| Genetic reagent (virus) | AAV-EF1α-DIO-EGFP-2A-TeNT | Thomas Südhof Lab at Stanford University | NA | |
| Genetic reagent (virus) | pAAV-EF1α-DIO-ChR2-mCherry | Addgene | RRID:Addgene_20297 | |
| Genetic reagent (virus) | AAV-EF1α-DIO-jGCaMP7s | Addgene | RRID:Addgene_104463 | |
| Genetic reagent (virus) | AAV2-retro-hSyn-DIO-EGFP | TaiTool | NA | |

*Continued on next page*

*Continued*

| Reagent type (species) or resource | Designation | Source or reference | Identifiers | Additional information |
|---|---|---|---|---|
| Genetic reagent (virus) | AAV2/9-hSyn-DIO-EGFP-2A-TeNT | TaiTool | NA | |
| Genetic reagent (virus) | AAV2/9-hSyn-DIO-EGFP | TaiTool | NA | |
| Genetic reagent (virus) | AAV2/9-hSyn-DIO-ChR2-mCherry | TaiTool | NA | |
| Genetic reagent (virus) | AAV2/9-hSyn-DIO-mCherry | TaiTool | NA | |
| Genetic reagent (virus) | AAV2/9-hSyn-DIO-jGCaMP7s | TaiTool | NA | |
| Genetic reagent (virus) | AAV2/9-EF1α-DIO-EGFP-2A-TVA | BrainVTA | NA | |
| Genetic reagent (virus) | AAV2/9-EF1α-DIO-RV-G | BrainVTA | NA | |
| Genetic reagent (virus) | RV-EnvA-ΔG-DsRed | BrainVTA | NA | |
| Antibody | Rabbit polyclonal anti-EGFP | Abcam | Cat# ab290, RRID:AB_303395 | |
| Antibody | Polycolonal anti-mCherry | Abcam | Cat# ab167453, RRID:AB_2571870; Cat# ab205402, RRID:AB_2722769 | |
| Sequence-based reagent | M-Cbln2-cre-upF | Tsingke Biological Technology, China | Primer | 5'-GGTACCTACTGT GTATCGCCAG-3' |
| Sequence-based reagent | CRE-AS | Tsingke Biological Technology, China | Primer | 5'-CTGTTTCACTAT CCAGGTTACG-3' |
| Sequence-based reagent | CRE-S | Tsingke Biological Technology, China | Primer | 5'-TACTGACGGT GGGAGAATG-3' |
| Sequence-based reagent | M-Cbln2-ires-cre-doR | Tsingke Biological Technology, China | Primer | 5'-GTTTGAAGCTG CACTGAGAGAG-3' |
| Chemical compound, drug | D-AP5/CNQX | Tocris | Cat# 0106 / 0190 | |
| Chemical compound, drug | Picrotoxin/TTX | Tocris | Cat# 1128 / 1078 | |
| Chemical compound, drug | 4-AP | Sigma | Cat# 275875 | |
| Chemical compound, drug | DAPI | Sigma | Cat# D8417 | |
| Software, algorithm | GraphPad Prism 9.0.0 | *GraphPad, 2015* | https://www.graphpad.com/scientific-software/prism/ | |
| Software, algorithm | Cell ranger 3.0.2 | *Zheng et al., 2017* | http://10xgenomics.com | |
| Software, algorithm | R version 3.6.1 | *R Development Core Team, 2020* | https://www.r-project.org | |
| Software, algorithm | Seurat 3.1.0 | *Stuart et al., 2019* | https://satijalab.org/seurat/ | |
| Software, algorithm | Scrublet 0.2.1 | *Wolock et al., 2019* | https://github.com/swolock/scrublet | |
| Software, algorithm | batchelor 1.0.1 | *Haghverdi et al., 2018* | https://github.com/MarioniLab/MNN2017/ | |
| Software, algorithm | pheatmap 1.0.12 | *Kolde, 2019* | https://github.com/raivokolde/pheatmap | |
| Software, algorithm | rgl 0.100.54 | *Adler and Murdoch, 2020* | https://github.com/dmurdoch/rgl | |
| Software, algorithm | metascape | *Zhou et al., 2019* | https://metascape.org/gp/index.html#/main/step1 | |

*Continued on next page*

*Continued*

| Reagent type (species) or resource | Designation | Source or reference | Identifiers | Additional information |
|---|---|---|---|---|
| Software, algorithm | Image J v1.48h3 | *Schneider et al., 2012* | https://imagej.nih.gov/ij/ | |
| Software, algorithm | MATLAB 2019b | *MATLAB, 2018* | https://www.mathworks.com | |
| Software, algorithm | Brainrender 2.0 | *Claudi et al., 2020* | https://github.com/brainglobe/brainrender | |

## Mice

All experimental procedures were conducted following protocols approved by the Administrative Panel on Laboratory Animal Care at the National Institute of Biological Sciences, Beijing (NIBS) (NIBS2021M0006) and Institute of Biophysics, Chinese Academy of Sciences (SYXK2019015). The *Pitx2-Cre* knock-in line (*Liu et al., 2003*) was imported from the Mutant Mouse Resource Centers (MMRRC_000126-UCD). The vGlut2-IRES-Cre (also called *Slc17a6-IRES-Cre*) mice (*Vong et al., 2011*) were imported from the Jackson Laboratory (JAX Mice and Services). Mice were maintained on a circadian 12 hr light/12 hr dark cycle with food and water available ad libitum. Mice were housed in groups (three to five animals per cage) before they were separated 3 days prior to virus injection. After virus injection, each mouse was housed in one cage for 3 weeks before subsequent experiments. To avoid potential sex-specific differences, we used male mice only.

## Nuclei preparation

Mice were anesthetized with 3% isoflurane and brains were removed and placed into ice-cold oxygenated artificial cerebrospinal fluid (aCSF). SC were dissected from the midbrain and placed into RNAlater (Invitrogen, AM7021) and stored at 4°C overnight. To ensure the quality of the experiment, two replicates were conducted and five mice were used for each replicate. On the day for the experiment, tissue samples were washed with phosphate buffered saline (PBS) (Gibco, REF 10010–023) and cut into pieces <1 mm and were homogenized using a glass Dounce tissue grinder (Sigma, Cat# D8938) in 2 ml of ice-cold EZ PREP (Sigma, Cat# NUC-101). Then the nuclei suspension was transferred into a 15 ml tube and incubated on ice for 5 min with 2 ml of ice-cold EZ PREP added. After incubation, the nuclei were centrifuged at $500\times$ *g* for 5 min at 4°C. The nuclei were re-suspended with 4 ml ice-cold EZ PREP and incubated on ice for another 5 min. Then the nuclei were centrifuged at $500\times$ *g* for 5 min at 4°C and washed in 4 ml Nuclei Suspension Buffer (NSB; consisting of $1\times$ PBS, 0.04% BSA, and 0.1% RNase inhibitor [Clontech, Cat# 2313A]). After being re-suspended in 2 ml NSB, the nuclei were filtered with a 35 μm cell strainer (Corning, Cat# 352235). The nuclei density was adjusted to 1,000,000 nuclei/ml and placed on ice for use.

## snRNA-seq library construction

Libraries were prepared using $10\times$ GENOMICS platform following the RNA library preparation protocols. Briefly, by using the $10\times$ GemCode Technology, thousands of nuclei were partitioned into nanoliter-scale Gel BeadIn-EMulsions (GEMs). At this step, all the cDNA produced from the same nuclei were labeled by a common $10\times$ Barcode. Primers containing an Illumina R1 sequence (read1 sequencing primer), a 16 bp $10\times$ Barcode, a 10 bp randomer and a poly-dT primer sequence were released and mixed with nuclei lysate and Master Mix upon dissolution of the single-cell 3′ gel bead in a GEM. The GEMs were incubated to generate barcoded, full-length cDNA from poly-adenylated mRNA by reverse transcription. After breaking the GEMs, silane magnetic beads were used to remove the leftover biochemical reagents and primers. Before constructing the library, the cDNA amplicon size was optimized by enzymatic fragmentation and size selection. During the end repair and adaptor ligation step, P5, P7, a sample index and R2 (read two primer sequence) were added to each selected cDNA. P5 and P7 primers were used in Illumina bridge amplification of the cDNA (http://10xgenomics.com). The libraries were sequenced using the Illumina HiSeq4000 with150 bp paired-end reads.

## High-throughput snRNA-seq data preprocessing and analyzing

For 10× snRNA-seq data, the reads were aligned to mouse reference genome mm10 with Cell Ranger (version 3.0.2) (*Zheng et al., 2017*). To detect potential doublets, we performed the scrublet (version 0.2.1) pipeline on each sample with parameters (expected_doublet_score=0.06, sim_doublet_ratio=20, min_gene_variability_pctl=85 and n_prin_comps=30) (*Wolock et al., 2019*). 1708/17979 cells with computed doublet score greater than 0.16 were identified as doublets and excluded from subsequent analysis. Next, a series of quality control analyses were performed. Cells with nGenes (number of detected genes) below 800 or above 6000 were discarded. Cells with nUMI (number of unique molecular identifier) above 20,000 or percentage of mitochondrial genes greater than 3% were removed. Genes that did not show expression in at least three cells were excluded. After quality control, 14,892 cells and 23,076 genes were kept for downstream analysis.

The downstream analysis of 10× snRNA-seq data was performed with R package Seurat (3.1.0) (*Butler et al., 2018*; *Stuart et al., 2019*). Briefly, a Seurat object was created with the filtered read counts. The log-transformation was then performed with the function NormalizeData. Next, 2000 variable genes were identified with function FindVariableGenes and passed to function RunPCA for the principal component analysis (PCA). Then, batch effect correction was performed using function fastMNN *Haghverdi et al., 2018* followed by dimension reduction with t-distributed stochastic neighbor embedding (t-SNE) approach using function RunTSNE. Subsequently, clustering analysis was performed with function FindClusters by setting parameter resolution to 2.0. Known markers *Slc17a6*, *Gad1*, *Mbp, Pdgfra*, *Aldh1a1*, *Cx3cr1*, *Cldn5*, *Foxc1*, and *Ccdc146* were used to name the major cell types excitatory neurons, inhibitory neurons, oligodendrocytes, OPCs, astrocytes, microglia cells, endothelial cells, meninges, and ciliated cells, respectively. In addition, we further subclustered excitatory neurons and inhibitory neurons into 9 and 10 subclusters, respectively, following the same procedure described above.

## Allen brain in situ data processing and layer specificity score calculation with computational method (SPACED)

To determine whether different SC neuronal subtypes own spatial layer specificity, we analyzed the in situ hybridization images of genes that are subtype specific from Allen Mouse Brain Atlas (https://mouse.brain-map.org/search/index) followed the computational method SPACED. Briefly, the DEGs of excitatory neuron and inhibitory neuron subtypes were firstly computed. To access genes whose expression pattern is more subtype restricted, we computed subtype specificity score for each gene based on Jensen-Shannon divergence, inspired by Cusanovich's study (*Cusanovich et al., 2018*) and ranked the DEGs by their subtype specificity score. Then, for each neuron subtype, the top 10 most subtype specific genes were selected as reference for spatial classification, as SPACED exhibited higher sensitivity with 10 genes (detailed information could be found at the Github repository for SPACED; https://github.com/xiaoqunwang-lab/SPACED). The in situ slices of the selected genes used for spatial classification were downloaded from Allen Mouse Brain Atlas (https://mouse.brain-map.org/search/index). After that, the color type of these slices was transformed into 8-bit using ImageJ (v1.48h3). The signal pixels of each slice were converted into red by performing 'Image>Adjust>Threshold' in ImageJ. Subsequently, the four layers of the SC, named as SuG layer, Op layer, InG layer/InWh layer, and DpG layer were selected using ROI manager. In each slice, a region with the weakest signal was selected as the background. The signal intensity of the four layers was calculated respectively. Briefly, the area fraction (definition from ImageJ: The percentage of pixels in the image or selection that have been highlighted in red using Image>Adjust>Threshold. For non-thresholded images, the percentage of non-zero pixels.) of each of the five ROIs in each slice was firstly calculated. The signal intensity was then carried out by subtracting the area fraction of the background ROI (ROI with the weakest signal) from that of the other four ROIs (SuG, Op, InG/InWh, and DpG). Afterward, the computed in situ signal intensities of each gene for the four SC layers were normalized into range 0 and 1. We then computed a log-transformation of the mean of signal intensity for each layer across the selected genes as the layer specificity score for the corresponding layer. To access the spatial distribution priority of each subtype, ANOVA and post hoc test were performed on the processed signal intensities and p-value < 0.05 was considered as statistically significant. Source code for the computational method SPACED is available at https://github.com/xiaoqunwang-lab/SPACED (*Wang, 2021a*).

## Comparison of neuronal subtypes and spatial mapping results

To compare the SC neuronal subtypes and their spatial mapping results identified in our study to those reported in previous work from *Zeisel et al., 2018*, we performed integration pipeline introduced by R package Seurat on these two datasets. Indeed, we first extracted excitatory neurons and inhibitory neurons from Zeisel's study with the assigned regional identities of SC by their spatial mapping results. To compare the neuronal subtypes and their mapping results proposed in two studies, we mapped our data onto Zeisel's data with Seurat functions FindIntegrationAnchors (*Butler et al., 2018*) and IntegrateData (*Stuart et al., 2019*). After dataset integration, we then performed dimension reduction and clustering analysis on the assembled dataset with functions RunU-MAP and FindClusters. Alluvial plots were then constructed for visualizing the mapping results for neuronal subtypes and regional identities reported in this study and Zeisel's study.

## Slice preparation and cell harvesting using patch-seq

After anesthetized with 3% isoflurane, the mice were decapitated and the brains were removed and placed into ice-cold oxygenated sucrose-based artificial cerebrospinal fluid (sucrose-aCSF) containing (in mM): 234 sucrose, 2.5 KCl, 1.25 NaH$_2$PO$_4$, 10 MgSO$_4$, 0.5, CaCl$_2$, 26 NaHCO$_3$, and 11 D-glucose, pH 7.4. Brains were cut into 200-µm-thick slices in ice-cold oxygenated sucrose-aCSF with a microtome (Leica VT 1200S). Then the slices were incubated in oxygenated aCSF containing (in mM): 126 NaCl, 3 KCl, 26 NaHCO$_3$, 1.2 NaH$_2$PO$_4$, 10 D-glucose, 2.4 CaCl$_2$, and 1.3 MgCl$_2$, pH 7.4 at room temperature for 1 hr. To pick the fluorescence labeled neurons in SC, glass capillaries (2.0 mm OD, 1.16 mm ID, Sutter Instruments) were autoclaved prior to pulling patch-seq pipettes. All the surfaces of the environment were kept clean and RNase-free with DNA-OFF (Takara Cat# 9036) and RNase Zap (Life Technologies Cat# AM9780). To ensure a successful harvest of the cell, the patch-seq pipettes were pulled by a micropipette puller (Sutter Instrument, MODEL P-97) until the resistance was 2–4 MΩ. The pipette solution containing 123 mM potassium gluconate, 12 mM KCl, 10 mM HEPES, 0.2 mM EGTA, 4 mM MgATP, 0.3 mM NaGTP, 10 mM sodium phosphocreatine, 20 µg/ml glycogen, and 1 U/µl recombinant RNase inhibitor (Takara Cat# 2313A), pH ~7.25 was prepared. Cells were absorbed into patch-seq pipettes filled with pipette solution and ejected into RNase-free PCR tube containing 4 µl of RNase-free lysis buffer consisting of: 0.1% Triton X-100, 5 mM (each) dNTPs, 2.5 µM Oligo-dT30, 1 U/µl RNase inhibitor, and ERCC RNA Spike-In Mix (Life Technologies Cat# 4456740) (*Cadwell et al., 2016*).

## Patch-seq library construction and sequencing

The RNA collected from neurons by patch-seq were converted to DNA with the Smart-seq2 protocol (*Picelli et al., 2014*). Briefly, reverse transcription of the poly(A)-tailed mRNA with SuperScript II reverse transcriptase (Invitrogen REF 18064–014) was carried out. After 20 cycles of amplification, about 50–100 ng cDNA were produced; 25 ng cDNA was used as input DNA to construct the library with KAPA HyperPlus Kit (KAPABIOSYSTEM, KK8514). Briefly, the cDNA were fragmented with fragmentation enzymes for 20 min at 37°C. Then the fragmented cDNA were proceeded to end repair and A-tailing at 65°C for 30 min. After adaptor ligation step, the cDNA were amplified with six to eight cycles to produce enough library DNA for sequencing. The libraries were sequenced using Illumina HiSeq 2000.

## Patch-seq data preprocessing and analyzing

Adapter and low-quality reads were discarded with Python script AfterQC (*Chen et al., 2017*). Paired-end reads were aligned to the mouse reference genome mm10 using software STAR (STAR 2.5.3a) (*Dobin et al., 2013*) with default parameters except for the use of setting output type (–out-SAMtype). Reads were then counted with featureCounts (featureCounts 1.5.3) (*Liao et al., 2014*). Cells with nGene between 200 and 10,000, percentage of mitochondrial genes lower than 10%, and percentage of ERCC below 5% were included. Genes that have expression in at least two cells were included. The filtered counts contained then 60 cells and 19,541 genes. The downstream analysis for Smart-seq data was carried out with R package Seurat (*Butler et al., 2018*; *Stuart et al., 2019*). Gene expression normalization was performed with function NormalizeData followed by computing variable genes using function FindVariableGenes. For dimensionality reduction, PCA and t-SNE

approaches were applied with functions RunPCA and RunTSNE, respectively. Clustering analysis was done with FindClusters function by setting resolution to 1.

## Mapping patch-seq data on high-throughput snRNA-seq data

To mapping Smart-seq clusters onto 10× clusters, we first performed CCA alignment with Seurat functions FindIntegrationAnchors and IntegrateData (*Butler et al., 2018*; *Stuart et al., 2019*) on these two datasets to remove potential technical batch effect. We then computed the correlation coefficient between Smart-seq and 10× clusters based on the CCA integrated data.

## Identification of DEGs

For 10× high-throughput snRNA data, DEGs were computed using FindAllMarkers function (*Butler et al., 2018*; *Stuart et al., 2019*) with method Wilcox. Genes with adjusted $P_{adj}$ < 0.05 were identified as DEGs. For Smart-seq data, DEGs were computed using FindAllMarkers function with method roc. Genes with a power > 0.4 were identified as DEGs.

## Tissue preparation and two-photon imaging

In AAV tracing experiments, brains were harvested 4 weeks after viral injection, post-fixed in 4% paraformaldehyde (PFA) at 4°C overnight (12–14 hr), rinsed in phosphate buffered saline for 15 min three times, and sliced into series of 120-µm-thick coronal sections with a vibratome (Leica VT1200S, Leica). Complete tissue sections were scanned using 25× water-immersion objectives on a two-photon microscope (Nikon). Sections were imaged with 920 nm excitation wavelengths. Z-series images were taken at 2 µm steps. Threshold parameters were individually adjusted for each case using the ImageJ (v1.53c).

## M-CRITIC

The dendrites and/or axons were traced using the ImageJ plug-in Simple Neurite Tracer (semiautomatic tracing) and the tracing results were saved in SWC format. Full anatomical morphology of individual neuron was reconstructed from a serial of aligned image-tracing stacks by manual works and custom-written MATLAB (MathWorks, Natick, MA, R2019b) program (https://github.com/xiaoqun-wang-lab/M-CRITIC). Subsequently, reconstructions of neuron morphology were registered to the Allen Mouse CCF (*Wang et al., 2020b*). Two experienced individuals performed back-to-back manual validation of the registration results.

## Generation of Cbln2-IRES-Cre mice

The Cbln2-IRES-Cre mice were produced using CRISPR/Cas9 system based on the method described before (*Ma et al., 2017*). In brief, two sgRNA targeting sites A (Sequence: GGAGAAGA-GAACAGAAGGTG) and B (Sequence: GAGCCACCAGGATGATGGGA) were used for Cbln2 targeting. All homologous recombination donor templates were prepared on the basis of the mice genomic sequence (AssemblyGRCm38.p6) by insertion of IRES-Cre sequence to the end of each targeting gene. The transcribed sgRNA and purified donor templates were mixed with Cas9 protein for mice embryo microinjection. The newborn pup genomic DNA was extracted from 7-day-old mice tail based on the method described before (*Ma et al., 2017*). Genotyping was performed using primers listed in Key resources table. The correct insertion was further confirmed by sequencing.

## AAV vectors

The AAV serotype used in the present study is AAV2/9. The AAVs used in the present study are listed in Key resources table. AAV-EF1α-DIO-EGFP-2A-TeNT was from Thomas Südhof Lab at Stanford University. The plasmid for pAAV-EF1α-DIO-ChR2-mCherry (Addgene #20297) was from Deisseroth Lab. The cDNA for AAV-EF1α-DIO-jGCaMP7s was from Kim Lab (Addgene #104463). The viral particles were prepared by Taitool Inc and BrainVTA Inc. The produced viral vector titers before dilution were in the range of 0.8–1.5×$10^{13}$ viral particles/ml. The final titer used for AAV injection is 5×$10^{12}$ viral particles/ml. The AAV mixture for sparse labeling was produced in Minmin Luo's Laboratory (*Lin et al., 2018*). The titer of AAV-TRE-DIO-Flpo was 1×$10^{10}$ particles/ml. The titer of AAV-TRE-fDIO-GFP-IRES-tTA was 1×$10^{12}$ particles/ml. The ratio for mixture was 1:9.

## Stereotaxic injection

Mice were anesthetized with an intraperitoneal injection of tribromoethanol (125–250 mg/kg). Standard surgery was performed to expose the brain surface above the SC, ZI, and LPTN. Coordinates used for SC injection were: bregma −3.80 mm, lateral ±1.00 mm, and dura −1.25 mm. Coordinates used for ZI injection were: bregma −2.06 mm, lateral ±1.25 mm, and dura −4.00 mm. Coordinates used for LPTN injection were: bregma −2.30 mm, lateral ±1.50 mm, and dura −2.30 mm. The injection was performed with the pipette connected to a Nano-liter Injector 201 (World Precision Instruments, Inc) at a slow flow rate of 0.15 µl/min to avoid potential damage to local brain tissue. The pipette was withdrawn at least 20 min after viral injection. For optogenetic activation and fiber photometry experiments, AAV injections were unilateral and were followed by ipsilateral optical fiber implantation (see 'Optical fiber implantation'). For TeNT-mediated synaptic inactivation experiments, AAV injections were bilateral.

## Optical fiber implantation

Thirty minutes after the AAV injection, a ceramic ferrule with an optical fiber (230 µm in diameter, NA 0.37) was implanted with the fiber tip on top of the Cbln2+ SC neurons (bregma −3.80 mm, lateral +0.75 mm, and dura −1.00 mm) or Pitx2+ SC neurons (bregma −3.80 mm, lateral +1.75 mm, and dura −1.75 mm). In some cases, the optical fiber was implanted with the fiber tip on top of the ZI (bregma −2.06 mm, lateral +1.25 mm, dura −4.00 mm) or LPTN (bregma −2.30 mm, lateral +1.50 mm, dura −2.30 mm). The ferrule was then secured on the skull with dental cement. After implantation, the skin was sutured, and antibiotics were applied to the surgical wound. The optogenetic and fiber photometry experiments were conducted at least 3 weeks after optical fiber implantation. All experimental designs related to optical fiber implantation are summarized in *Supplementary file 6*. For optogenetic stimulation, the output of the laser was measured and adjusted to 5, 10, 15, and 20 mW before each experiment. The pulse onset, duration, and frequency of light stimulation were controlled by a programmable pulse generator attached to the laser system. After AAV injection and fiber implantation, the mice were housed individually for 3 weeks before the behavioral tests.

## Preparation of the behavioral tests

Before the behavioral tests, the animals were handled daily by the experimenters for at least 3 days. On the day of the behavioral test, the animals were transferred to the testing room and were habituated to the room conditions for 3 hr before the experiments started. The apparatus was cleaned with 20% ethanol to eliminate odor cues from other animals. All behavioral tests were conducted during the same circadian period (13:00–19:00). All behaviors were scored by the experimenters, who were blind to the animal treatments.

## Visually evoked freezing response

Visually evoked freezing response was measured according to the established behavioral paradigm in a standard arena (35 cm × 35 cm square open field) with regular mouse bedding. A regular computer monitor was positioned above the arena for presentation of overhead moving visual target. After entering, the mice were allowed to explore the arena for 10 min. This was followed by the presentation of a small visual target moving overhead. The visual target was a black circle (2.5 cm in diameter), which was 5° of visual angle, moving in a linear trajectory at 10 cm/s from one corner to the other of the monitor. The luminance of the black circle and the gray background was 0.1 and 3.6 cd/m$^2$, respectively. Mouse behavior was recorded (25 fps) by two orthogonally positioned cameras with LEDs providing infrared illumination. The location of the mouse in the arena (X, Y) was measured by a custom-written MATLAB program described previously (*Shang et al., 2018*). The instantaneous locomotion speed was calculated with a 200 ms time-bin. To quantitatively measure the freezing response, we calculated the average locomotion speed before (3 s), average speed during (5 s), and average speed after (5 s) visual stimuli.

For testing optogenetically evoked freezing response, a 473 nm diode pumped solid state laser system was used to generate the 473 nm blue laser for light stimulation. An FC/PC adaptor was used to connect the output of the laser to the implanted ferrule for intracranial light delivery. The mice were handled daily with all optics connected for at least 3 consecutive days before the

behavioral test to reduce stress and anxiety. Before each experiment, the output of the laser was adjusted to 5 mW. The pulse onset, duration, and frequency of light stimulation were controlled by a programmable pulse generator attached to the laser system. Locomotor behaviors before, during, and after light stimulation (10 Hz, 20 ms, 5 mW, 5 s) were recorded with two orthogonally positioned cameras and were measured by a custom-written MATLAB program described previously (*Shang et al., 2018*).

## Behavioral paradigm for predatory hunting

The procedure of predatory hunting experiment was described previously (*Shang et al., 2019*). Before the predatory hunting test, the mice went through a 9-day habituation procedure (days H1–H9). On each of the first 3 habituation days (days H1, H2, H3), three cockroaches were placed in the home-cage (with standard chow) of mice at 2:00 PM. The mice readily consumed the cockroaches within 3 hr after cockroach appearance. On days H3, H5, H7, and H9, we initiated 24 hr food deprivation at 7:00 PM by removing chow from the home-cage. On days H4, H6, and H8 at 5:00 PM, we let the mice freely explore the arena (25 cm × 25 cm) for 10 min, followed by three trials of hunting practice for the cockroach. After hunting practice, we put the mice back in their home-cages and returned the chow at 7:00 PM. On the test day, we let the mice freely explore the arena for 10 min, followed by three trials of predatory hunting. After the tests, the mice were put back in their home-cage, followed by the return of chow. The cockroach was purchased from a merchant in Tao-Bao Online Stores (http://www.taobao.com).

Before the hunting practice or test, the mice were transferred to the testing room and habituated to the room conditions for 3 hr before the experiments started. The arena was cleaned with 20% ethanol to eliminate odor cues from other mice. All behaviors were scored by the experimenters, who were blind to the animal treatments. Hunting behaviors were measured in an arena (25 cm × 25 cm, square open field) without regular mouse bedding. After entering, the mice explored the arena for 10 min, followed by the introduction of a cockroach. For each mouse, predatory hunting was repeated for three trials. Each trial began with the introduction of prey to the arena. The trial ended when the predator finished ingesting the captured prey. After the mice finished ingesting the prey body, debris was removed before the new trial began.

## Measurement of predatory attack in predatory hunting

In the paradigm of predatory hunting, mouse behavior was recorded in the arena with three orthogonally positioned cameras (50 frames/s; Point Grey Research, Canada). With the video taken by the overhead camera, the instantaneous head orientation of predator relative to prey (azimuth angle) and predator-prey distance was analyzed with the Software EthoVision XT 14 (Noldus Information Technology). With the videos taken by the two horizontal cameras, predatory attacks with jaw were visually identified by replaying the video frame by frame (50 frames/s). We marked the predatory jaw attacks with yellow vertical lines in the behavioral ethogram of predatory hunting. With this method, we measured three parameters of predatory hunting: time to capture, latency to attack, and attack frequency. Time to capture was defined as the time between the introduction of prey and the last jaw attack. Latency to attack was defined as the time between the introduction of the prey and the first jaw attack from the predator. Attack frequency was defined as the number of jaw attacks divided by time to prey capture. Data for three trials were averaged.

## Fiber photometry recording

A fiber photometry system (ThinkerTech, Nanjing, China) was used for recording GCaMP signals from genetically identified neurons. To induce fluorescence signals, a laser beam from a laser tube (488 nm) was reflected by a dichroic mirror, focused by a 10× lens (NA 0.3) and coupled to an optical commutator. A 2 m optical fiber (230 μm in diameter, NA 0.37) guided the light between the commutator and implanted optical fiber. To minimize photo bleaching, the power intensity at the fiber tip was adjusted to 0.02 mW. The GCaMP7s (*Dana et al., 2019*) fluorescence was band-pass filtered (MF525-39, Thorlabs) and collected by a photomultiplier tube (R3896, Hamamatsu). An amplifier (C7319, Hamamatsu) was used to convert the photomultiplier tube current output to voltage signals, which were further filtered through a low-pass filter (40 Hz cut-off; Brownlee 440). The

analogue voltage signals were digitalized at 100 Hz and recorded by a Power 1401 digitizer and Spike2 software (CED, Cambridge, UK).

AAV-hSyn-DIO-jGCaMP7s was stereotaxically injected into the SC of *Cbln2-IRES-Cre* mice or *Pitx2-Cre* mice followed by optical fiber implantation above the Cbln2+ SC neurons or Pitx2+ SC neurons (see 'Stereotaxic injection' and 'Optical fiber implantation'). Three weeks after AAV injection, fiber photometry was used to record GCaMP signals from the Cbln2+ SC neurons or Pitx2+ SC neurons of head-fixed mice standing on a treadmill in response to visual and vibrissal somatosensory stimuli (see below). A flashing LED triggered by a 1 s square-wave pulse was simultaneously recorded to synchronize the video and GCaMP signals. After the experiments, the optical fiber tip sites in the SC were histologically examined in each mouse.

### Visual stimulation

The test mice were head-fixed and standing on top of a cylindrical treadmill (Nanjing Thinktech Inc) for fiber photometry recording of SC neurons in response to visual stimuli. The contralateral eye was kept open, and the ipsilateral eye was covered to prevent viewing. A 45 cm wide and 35 cm high screen was placed 18 cm from the contralateral eye and 25° to the mid-sagittal plane of the mouse, resulting in a visually stimulated area (100° horizontal × 90° vertical) in the lateral visual field. The orientation of the screen was adjusted ~45° to make the screen perpendicular to the eye axis of the contralateral eye. After identification of the RF location on the screen of SC neurons, a computer-generated black circle (diameter = 5° or 25°) moving across the visual RF in eight direction at different speed (32°/s or 128°/s) was presented. The luminance of the black circle and gray background was 0.1 and 6.6 cd/m$^2$, respectively. The black circle first appeared stationary outside the RF for 2 s to collect baseline calcium signals as controls, and was then presented with an interval of at least 15 s between trials to allow the neurons to recover from any motion adaptation.

### Vibrissal air puff stimulation

To mimic the somatosensory cues of moving prey, brief air puffs (50 ms) with different strengths (15 or 30 psi) were delivered through a metal tube (diameter 1.5 mm) connected with Picospritzer III. The output of Picospritzer III was controlled by a programmable pulse generator. When delivering air puffs as vibrissal somatosensory stimuli, the tube was oriented from temporal to nasal side of mouse. The distance between the tube nozzle and the whiskers was ~30 mm. When presenting repetitive air puff stimuli, the frequency was either 0.5 or 2 Hz. For each mouse, 10–15 trials were repeatedly presented to the whiskers, so that an average response was obtained.

### Cell-type-specific anterograde tracing

For cell-type-specific anterograde tracing of Cbln2+ and Pitx2+ SC neurons, AAV-DIO-EGFP (200 nl) was stereotaxically injected into the SC of Cbln2-IRES-Cre and Pitx2-Cre mice, respectively. The mice were then maintained in a cage individually. Three weeks after viral injection, mice were perfused with saline followed by 4% PFA in PBS. After 8 hr of post-fixation in 4% PFA, coronal or sagittal brain sections at 40 μm in thickness were prepared using a cryostat (Leica CM1900). All coronal sections were collected and stained with primary antibody against EGFP and DAPI. The coronal brain sections were imaged with an Olympus VS120 epifluorescence microscope (10× objective lens).

### Cell-type-specific RV tracing

The modified RV-based three-virus system was used for mapping the whole-brain inputs to vGAT+ AHN neurons (*Wickersham et al., 2007*). All the viruses included AAV2/9-CAG-DIO-EGFP-2A-TVA (5 × 10$^{12}$ viral particles/ml), AAV2/9-CAG-DIO-RG (5 × 10$^{12}$ viral particles/ml), and EnvA-pseudo-typed, glycoprotein (RG)-deleted and DsRed-expressing RV (RV-EvnA-DsRed, RV) (5.0 × 10$^8$ viral particles/ml), which were packaged and provided by BrainVTA Inc (Wuhan, China). A mixture of AAV2/9-CAG-DIO-EGFP-2A-TVA and AAV2/9-CAG-DIO-RG (1:1, 200 nl) was stereotaxically injected into the SC of Cbln2-IRES-Cre or Pitx2-Cre mice unilaterally. Two weeks after AAV helper injection, RV-EvnA-DsRed (300 nl) was injected into the same location in the SC of Cbln2-IRES-Cre or Pitx2-Cre mice in a biosafety level-2 lab facility. Starter neurons were characterized by the coexpression of DsRed and EGFP, which were restricted in the SC.

One week after injection of RV, mice were perfused with saline followed by 4% PFA in PBS. After 8 hr of post-fixation in 4% PFA, coronal brain sections at 40 μm in thickness were prepared using a cryostat (Leica CM1950). All coronal sections were collected and stained with DAPI. The coronal brain sections were imaged with an Olympus VS120 epifluorescence microscope (10× objective lens) and analyzed with ImageJ. For quantifications of subregions, boundaries were based on the Allen Institute's reference atlas. We selectively analyzed the retrogradely labeled dense areas. The fractional distribution of total cells labeled by RV was measured.

## Cell-counting strategies

Cell-counting strategies are summarized in *Supplementary file 7*. For counting cells in the SC, we collected coronal sections (40 μm) from bregma −3.08 mm to bregma −4.60 mm for each mouse. We acquired confocal images (20× objective, Zeiss LSM 780) followed by cell counting with ImageJ software. By combining fluorescent in situ hybridization and immunohistochemistry, we counted the number of Cbln2+ and Pitx2+ cells in the SC and calculated the percentages of Cbln2+ and Pitx2+ neurons in the neuronal population labeled by EGFP. To analyze monosynaptic inputs of Cbln2+ SC neurons, we counted DsRed+ cells in a series of brain areas. For the detailed information on the brain regions and cell counting strategy, see *Supplementary file 7*. We acquired fluorescent images (10× objective, Olympus) followed by cell counting with ImageJ software.

## Slice physiological recording

Slice physiological recording was performed according to the published work (*Liu et al., 2017*). Brain slices containing the SC were prepared from adult mice anesthetized with isoflurane before decapitation. Brains were rapidly removed and placed in ice-cold oxygenated (95% $O_2$ and 5% $CO_2$) cutting solution (228 mM sucrose, 11 mM glucose, 26 mM $NaHCO_3$, 1 mM $NaH_2PO_4$, 2.5 mM KCl, 7 mM $MgSO_4$, and 0.5 mM $CaCl_2$). Coronal brain slices (400 μm) were cut using a vibratome (VT 1200S, Leica Microsystems). The slices were incubated at 28°C in oxygenated aCSF (119 mM NaCl, 2.5 mM KCl, 1 mM $NaH_2PO_4$, 1.3 mM $MgSO_4$, 26 mM $NaHCO_3$, 10 mM glucose, and 2.5 mM $CaCl_2$) for 30 min, and were then kept at room temperature under the same conditions for 1 hr before transfer to the recording chamber at room temperature. The aCSF was perfused at 1 ml/min. The acute brain slices were visualized with a 40× Olympus water immersion lens, differential interference contrast optics (Olympus Inc, Japan), and a CCD camera.

Patch pipettes were pulled from borosilicate glass capillary tubes (Cat# 64–0793, Warner Instruments, Hamden, CT) using a PC-10 pipette puller (Narishige Inc, Tokyo, Japan). For recording of action potentials (current clamp), pipettes were filled with solution (in mM: 135 K-methanesulfonate, 10 HEPES, 1 EGTA, 1 Na-GTP, 4 Mg-ATP, and 2% neurobiotin, pH 7.4). The resistance of pipettes varied between 3.0 and 3.5 MΩ. The voltage signals were recorded with MultiClamp 700B and Clampex 10 data acquisition software (Molecular Devices). After establishment of the whole-cell configuration and equilibration of the intracellular pipette solution with the cytoplasm, series resistance was compensated to 10–15 MΩ. Recordings with series resistances of >15 MΩ were rejected. An optical fiber (200 μm in diameter) was used to deliver light pulses, with the fiber tip positioned 500 μm above the brain slices. Laser power was adjusted to 5 mW. Light-evoked action potentials from ChR2-mCherry+ neurons in the SC were triggered by a light-pulse train (473 nm, 2 ms, 10 Hz or 20 Hz, 20 mW) synchronized with Clampex 10 data acquisition software (Molecular Devices).

## RNA in situ hybridization

Mice were perfused with PBS treated with 0.1% DEPC (Sigma, D5758), followed by DEPC-treated PBS containing 4% PFA (PBS-PFA). Brains were post-fixed in DEPC-treated PBS-PFA solution overnight and then placed in DEPC-treated 30% sucrose solution at 4°C for 30 hr. Brain sections to a thickness of 30 μm were prepared using a cryostat (Leica, CM3050S) and collected in DPEC-treated PBS. Fluorescence in situ hybridization (FISH) was performed as previously described (*Chen et al., 2020*) with minor modifications. Briefly, brain sections were rinsed with DPEC-treated PBS, permeabilized with DPEC-treated 0.1% Tween 20 solution (in PBS), and DPEC-treated 2 × SSC containing 0.5% Triton. Brain sections were then treated with $H_2O_2$ solution and acetic anhydride solution to reduce nonspecific FISH signals. After 2 hr incubation in prehybridization buffer (50% formamide, 5 × SSC, 0.1% Tween20, 0.1% CHAPS, 5 mM EDTA in DEPC-treated water) at 65°C, brain sections

were then hybridized with the hybridization solution containing mouse anti-sense cRNA probes (digoxigenin labeling) for Cbln2 (primers CAGCTTCCACGTGGTCAA and AGCCCCCAGCA TGAAAAC) or Pitx2 (primers CTCTCAGAGTATGTTTTCCCCG and AGGATGGGTCGTACATAGCAG T) at 65℃ for 20 hr. The sequences of cDNA primers for cRNA probes were the same as those in the ISH DATA of the Allen Brain Atlas (https://mouse.brain-map.org/). After washing, brain sections were incubated with Anti-Digoxigenin-POD, Fab fragments (1:400, Roche, 11207733910) at 4℃ for 30 hr, and FISH signals were detected using a TSA Plus Cyanine three kit (NEL744001KT, Perki-nElmer). To visualize the GFP signals, brain sections were incubated with a primary antibody against GFP (1:2000, ab290, Abcam) at 4℃ for 24 hr and then with an Alexa Fluor 488-conjugated goat anti-rabbit secondary antibody (1:500, A11034, Invitrogen) at room temperature for 2 hr. Brain sections were mounted and imaged using a Zeiss LSM780 confocal microscope or the Olympus VS120 Slide Scanning System.

## Immunohistochemistry

Mice were anesthetized with isoflurane and sequentially perfused with saline and PBS containing 4% PFA. Brains were removed and incubated in PBS containing 30% sucrose until they sank to the bottom. Post-fixation of the brain was avoided to optimize immunohistochemistry. Cryostat sections (40 μm) were collected, incubated overnight with blocking solution (PBS containing 10% goat serum and 0.7% Triton X-100), and then treated with primary antibodies diluted with blocking solution for 3–4 hr at room temperature. Primary antibodies used for immunohistochemistry are displayed in Key resources table. Primary antibodies were washed three times with washing buffer (PBS containing 0.7% Triton X-100) before incubation with secondary antibodies (tagged with Cy2, Cy3, or Cy5; dilution 1:500; Life Technologies Inc, Carlsbad, CA) for 1 hr at room temperature. Sections were then washed three times with washing buffer, stained with DAPI, and washed with PBS, transferred onto Super Frost slides, and mounted under glass coverslips with mounting media.

Sections were imaged with an Olympus VS120 epifluorescence microscope (10× objective lens) or a Zeiss LSM 710 confocal microscope (20× and 60× oil-immersion objective lens). Samples were excited by 488, 543, or 633 nm lasers in sequential acquisition mode to avoid signal leakage. Saturation was avoided by monitoring pixel intensity with Hi-Lo mode. Confocal images were analyzed with ImageJ software.

## Data quantification and statistical analyses

All experiments were performed with anonymized samples in which the experimenter was unaware of the experimental conditions of the mice. For the statistical analyses of experimental data, Student's t-test and one-way ANOVA were used. The 'n' used for these analyses represents number of mice or cells. See the detailed information of statistical analyses in figure legend and in *Supplementary file 8*. All statistical comparisons were conducted on data originating from three or more biologically independent experimental replicates. All data are shown as means ± SEM.

## Acknowledgements

We thank Karl Deisseroth for providing plasmids. We also thank the members of the Neuroscience Pioneer Club for valuable discussions. This work was supported by the National Key R and D Program of China (2019YFA0110100), the National Basic Research Program of China (2017YFA0102601, 2017YFA0103303), the Strategic Priority Research Program of the Chinese Academy of Sciences (XDB32010100), the National Natural Science Foundation of China (NSFC) (31925019 to PC and 31771140, 81891001 to X W), the BUAA-CCMU Big Data and Precision Medicine Advanced Innovation Center Project (BHME-2019001) to X W, and the institutional grants from the Chinese Ministry of Science and Technology to NIBS.

# Additional information

## Funding

| Funder | Grant reference number | Author |
|---|---|---|
| Ministry of Science and Technology of the People's Republic of China | 2019YFA0110100 | Xiaoqun Wang |
| Ministry of Science and Technology of the People's Republic of China | 2017YFA0102601 | Qian Wu |
| Ministry of Science and Technology of the People's Republic of China | 2017YFA0103303 | Xiaoqun Wang |
| Chinese Academy of Sciences | XDB32010100 | Xiaoqun Wang |
| National Natural Science Foundation of China | 31925019 | Peng Cao |
| National Natural Science Foundation of China | 31771140 | Xiaoqun Wang |
| National Natural Science Foundation of China | 81891001 | Xiaoqun Wang |
| BUAA-CCMU Big Data and Precision Medicine Advanced Innovation Center Project | BHME-2019001 | Xiaoqun Wang |

The funders had no role in study design, data collection and interpretation, or the decision to submit the work for publication.

## Author contributions

Zhiyong Xie, Zeyuan Liu, Congping Shang, Changjiang Zhang, Huating Gu, Gengxin Ran, Qing Pei, Qiang Ma, Meizhu Huang, Junjing Zhang, Jiyao Zhang, Miao Zhao, Investigation, Writing - review and editing; Mengdi Wang, Formal analysis, Investigation, Methodology, Writing - review and editing; Le Sun, Investigation, Methodology, Writing - review and editing; Rui Lin, Youtong Zhou, Minmin Luo, Resources, Writing - review and editing; Qian Wu, Peng Cao, Conceptualization, Supervision, Funding acquisition, Writing - original draft, Writing - review and editing; Xiaoqun Wang, Conceptualization, Supervision, Funding acquisition, Project administration, Writing - review and editing

## Author ORCIDs

Zhiyong Xie https://orcid.org/0000-0002-5691-3357
Mengdi Wang https://orcid.org/0000-0002-0214-2913
Zeyuan Liu https://orcid.org/0000-0003-0007-9874
Minmin Luo http://orcid.org/0000-0003-3535-6624
Qian Wu https://orcid.org/0000-0002-7469-1583
Peng Cao https://orcid.org/0000-0001-7739-6857
Xiaoqun Wang https://orcid.org/0000-0003-3440-2617

## Ethics

Animal experimentation: All experimental procedures were conducted following protocols approved by the Administrative Panel on Laboratory Animal Care at the National Institute of Biological Sciences, Beijing (NIBS) (NIBS2021M0006) and Institute of Biophysics, Chinese Academy of Sciences (SYXK2019015).

## Decision letter and Author response

Decision letter https://doi.org/10.7554/eLife.69825.sa1
Author response https://doi.org/10.7554/eLife.69825.sa2

# Additional files

## Supplementary files

- Supplementary file 1. Sample information for high-throughput single-nucleus RNA-sequencing (snRNA-seq) of nucleus from superior colliculus (SC) and top 50 differentially expressed genes among nine major cell types in SC, related to *Figure 1A*.

- Supplementary file 2. Differentially expressed genes among 9 excitatory neuron subtypes and 10 inhibitory neuron subtypes, related to *Figure 1C–D*.

- Supplementary file 3. Sample information for patch-seq of neurons from superior colliculus (SC), related to *Figure 2H*.

- Supplementary file 4. Differentially expressed genes between zona incerta (ZI)- and lateral posterior thalamic nucleus (LPTN)-projecting neurons in superior colliculus (SC), related to *Figure 2I*.

- Supplementary file 5. Differentially expressed genes between excitatory neuron subtype Ex-3 and Ex-6, and between Ex-1 and Ex-4, related to *Figure 2L–M*, respectively.

- Supplementary file 6. Summary of all experimental designs.

- Supplementary file 7. Summary of cell-counting strategies.

- Supplementary file 8. Summary of statistical analyses.

- Transparent reporting form

## Data availability

The scRNA-seq data used in this study have been deposited in the Gene Expression Omnibus (GEO) under accession numbers GSE162404 (https://www.ncbi.nlm.nih.gov/geo/query/acc.cgi?acc=GSE162404).

The following dataset was generated:

| Author(s) | Year | Dataset title | Dataset URL | Database and Identifier |
|---|---|---|---|---|
| Liu Z, Wang M, Wu Q, Wang X | 2020 | Revealing the molecular mechanism of mouse innate behavior by single-cell sequencing | https://www.ncbi.nlm.nih.gov/geo/query/acc.cgi?acc=GSE162404 | NCBI Gene Expression Omnibus, GSE162404 |

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
