## [Decision Letter]

**Acceptance summary:**

This excellent manuscript combines molecular, anatomical and behavioral methods to characterize neuron types in the mouse superior colliculus. It will likely be a significant resource to those who study how these circuits integrate sensory information to promote motor output. A diverse set of experiments supports the conclusion that the superior colliculus includes separate circuit modules involved in distinct behaviors: prey capture and predator escape.

**Decision letter after peer review:**

Thank you for submitting your article "Transcriptomic encoding of sensorimotor transformation in the midbrain" for consideration by *eLife*. Your article has been reviewed by 2 peer reviewers, and the evaluation has been overseen by Reviewing Editor Sacha Nelson and Ronald Calabrese as the Senior Editor. The following individual involved in review of your submission has agreed to reveal their identity: David Feldheim (Reviewer #1).

Essential revisions:

We have included the full reviews below so as to preserve the clarity of the requested changes. However the required changes are those involving new analyses and textual clarifications as requested by Reviewer # 2.

After consultation, there was agreement that additional suggested experiments, such as the additional manipulations and recordings suggested by reviewer #1 and the in situ experiments suggested by reviewer #2 would strengthen the manuscript, but are not required for acceptance.

Please do, however, complete each of the additional analyses and clarifications requested in items 1.1, 1.2, 1.3, 2.1, 2.3 and 3.2 as outlined by reviewer #2 and consider the suggestion made in item 3.1.

*Reviewer #1 (Recommendations for the authors):*

My only question is if the medial and laterally labeled Cbln2 neurons have similar properties. For example in FigS4, the receptive field mapping of Cbln2 Cre mice using Ca imaging, it looks like the field of SC being imaged is medial. Therefore it would be expected that these cells would detect dorsal stimuli based on their position in the SC. It would be interesting to see if the more lateral neurons also had dorsal RFs.

A similar question comes up when assaying the behaviors are medial SC neurons being selectively stimulated/ablated in these experiments.

*Reviewer #2 (Recommendations for the authors):*

1. Single-cell RNA sequencing data

1.1 There is a large fraction of excitatory cells that were not assigned to a layer ("other neuron" in figure 1H, mostly Ex-2). This is a large fraction of the excitatory neurons. The data presented here suggest at least two possibilities: either these cells are widely distributed in the superior colliculus, across layers, or these are low-quality cells. If there is any evidence that these are low-quality cells, that should be clearly indicated in the paper. It would be useful to show in the supplementary data quality control plots (for example, a tSNE with cells color-coded by number of genes/cell). Why there are no cells mapping to the deep gray layer?

1.2. Neurons from the superior colliculus were sequenced by Zeisel et al. (Cell 2018). That study reported a comparable number of excitatory and inhibitory clusters. How do these new data compare to the Zeisel data? A Sankey plot or some other analysis showing the correspondence between these two datasets will help readers connecting the two resources.

1.3. How does SPACED compare to previous methods for mapping single-cell transcriptomes on the Allen Brain Atlas data? For example, are the results of mapping with SPACED consistent with the results obtained with the mapping method proposed by Zeisel et al. 2018? How does the method perform if the number of genes selected for mapping changes? How was the method benchmarked?

2. Morphological reconstructions and Patch-Seq

2.1. For the data shown in Figure 2A-G, the text refers to the injection of an unspecified "AAV mixture". Please describe in the main text what mixture was injected and what is the rationale of the experiment

2.2. The integration of molecular and projection data relies entirely on the Patch-seq experiments. Figure 2J, however, shows that there are neurons retrogradely labeled from InG-ZI and Op-LPTN that do not express neither Cbln2 not Pitx2. Is it possible that these neurons belong to another excitatory type? In these experiments, retrograde tracing from LPTN and ZI is not cell type specific, therefore the existence of an excitatory type that projects to these two brain areas and does not express neither Cbln2 nor Pitx2 cannot be excluded. I suggest performing FISH for Cbln2 and Pitx2 on vGlut-IRES-Cre animals injected with AAV2-retro-DIO-EGFP in the LPTN or ZI.

2.3. Related to the previous point, the Methods mention that the Patch-seq data and the scRNAseq data are integrated using the CCA method in Seurat. However, here only a correlation matrix is shown (Figure 2K). A tSNE or UMAP showing the integrated scRNAseq and Patch-seq data would be much more informative.

3. Anterograde tracing

3.1 I would consider describing the experiments in Figure 6 earlier in the manuscript, after Figure 2. That seems the most logical place. Furthermore, figure S6 deserves to be part of one of the main figures. These are beautiful experiments showing very clearly that Cbln2 and Pitx2 neurons project to different brain areas. However, the narrative of the paper is prone to mislead the reader, because it emphasizes the projections to LPTN and ZI.

3.2 Related to the previous point, it would be helpful to expand the Discussion by mentioning how the different brain areas innervated by Cbln2 and Pitx2 neurons may participate to predator escape and prey capture behaviors.

4. Calcium imaging

It is good to see representative traces from the fiber photometry data. However, it would be even nicer to show a raster plot or a heat map encompassing all the neurons that were recorded. I could not find anywhere in the manuscript how many neurons were recorded and from how many animals. Information on the data analysis pipeline for these data is also missing (or at least I could not find it easily). All this information should be included in the manuscript.

---

## [Author Response]

Essential revisions:We have included the full reviews below so as to preserve the clarity of the requested changes. However the required changes are those involving new analyses and textual clarifications as requested by Reviewer # 2.After consultation, there was agreement that additional suggested experiments, such as the additional manipulations and recordings suggested by reviewer #1 and the in situ experiments suggested by reviewer #2 would strengthen the manuscript, but are not required for acceptance.Please do, however, complete each of the additional analyses and clarifications requested in items 1.1, 1.2, 1.3, 2.1, 2.3 and 3.2 as outlined by reviewer #2 and consider the suggestion made in item 3.1.Reviewer #1 (Recommendations for the authors):My only question is if the medial and laterally labeled Cbln2 neurons have similar properties. For example in FigS4, the receptive field mapping of Cbln2 Cre mice using Ca imaging, it looks like the field of SC being imaged is medial. Therefore it would be expected that these cells would detect dorsal stimuli based on their position in the SC. It would be interesting to see if the more lateral neurons also had dorsal RFs.

We thank the reviewer for pointing this issue out. In this study, we made fiber photometry recording primarily from the Cbln2+ neurons in the medial SC, because these neurons may be more involved in monitoring the visual targets in the upper visual field, as suggested in a recent review paper on the SC circuits (Ito and Feldheim, 2018). We agree with the reviewer that it would be interesting to see if the more lateral Cbln2+ SC neurons have dorsal or ventral receptive fields. Although we did not do this experiment in the present study, it is very likely that their visual receptive fields are more ventrally distributed in the lateral visual field. We will make a more detailed analyses of Cbln2+ SC neurons in our future study.

A similar question comes up when assaying the behaviors are medial SC neurons being selectively stimulated/ablated in these experiments.

We thank the reviewer for pointing this issue out. Indeed, when examining how photostimulation of Cbln2+ SC neurons evoked behaviors, we primarily stimulated the Cbln2+ neurons in the medial SC, because these neurons may be more involved in monitoring the visual targets in the upper visual field (Ito and Feldheim, 2018). When examining how inactivation of Cbln2+ SC neurons affected behaviors, we expressed TeNT in Cbln2+ SC neuron in both lateral and medial SC. In future study, we will systematically examine how Cbln2+ neurons in the medial and / or lateral SC participate in visually-evoked freezing in mice.

Reviewer #2 (Recommendations for the authors):1. Single-cell RNA sequencing data1.1 There is a large fraction of excitatory cells that were not assigned to a layer ("other neuron" in figure 1H, mostly Ex-2). This is a large fraction of the excitatory neurons. The data presented here suggest at least two possibilities: either these cells are widely distributed in the superior colliculus, across layers, or these are low-quality cells. If there is any evidence that these are low-quality cells, that should be clearly indicated in the paper. It would be useful to show in the supplementary data quality control plots (for example, a tSNE with cells color-coded by number of genes/cell).

We thank the reviewer for the comments.

To find out why excitatory neurons from Ex-2 were not assigned to any layers, we first checked if these cells are low quality cells. As shown in Author response image 1, cluster Ex-2 have shown comparable levels of number of genes (nGene) and unique molecular identifiers (nUMI) as other neuronal clusters. In addition, to explore if there is any impact of technical variation on the clustering analysis, we computed the correlation between the reduced PCA (Principal Component Analysis) components with sequencing depth. As shown in Author response image 1, there is no strong correlation between PCs and sequencing depth. Based on the above two analyses, we think the cell quality and technical variation (sequencing depth) were not the causes for the undetermined spatial classification of cells form excitatory neuron cluster Ex-2.

We then checked the spatial distribution of the cells with expression of genes enriched in Ex-2 (Author response image 1) and found that cells with expression of genes *Rasgrf2* and *Kcnh1* are more likely to be located in the deep gray layer, while cells with expression of genes *Rasgef1b* and *Foxp2* are more widely distributed across layers in the SC. Ex-2 showed relative higher specificity to the deep gray layer compared to other parts of SC (Figure 1G), but it didn’t pass the test of statistical significance with an adjusted p-value of 0.5389. So we think the cells from Ex-2 were relatively widely distributed in the SC.

**Author response image 1. sa2fig1:** Quality control metrices and spatial distribution of Ex-2. (**A**) Violin plots showing the number of detected genes (nGene) and unique molecular identifiers (uUMI) of each cell types identified in the SC. (**B**) Scatter plot displaying the correlation between computed reduced dimension components (PCs) and sequencing depth. (**C**) Spatial distribution of the DEGs enriched in Ex2. Upper panel: gene expression among excitatory neurons were projected onto the two-dimensional t-SNE. Cells were colored according to relative gene expression level (gray, low; red, high). Lower panel: In situ hybridization staining of mouse superior colliculus for the identified DEGs (from the Allen Brain Atlas).

Why there are no cells mapping to the deep gray layer?

The Figure 1G have shown the layer specificity of each neuronal clusters. Excitatory neuron cluster Ex-2 showed relative higher specificity to the deep gray layer of SC, but it didn’t pass the test of statistical significance with an adjusted p-value of 0.5389. Therefore, we conclude that there was no neuronal clusters mapped to the deep gray layer with statistical significance.

1.2. Neurons from the superior colliculus were sequenced by Zeisel et al. (Cell 2018). That study reported a comparable number of excitatory and inhibitory clusters. How do these new data compare to the Zeisel data? A Sankey plot or some other analysis showing the correspondence between these two datasets will help readers connecting the two resources.

We thank the reviewer for this insightful suggestion. In Zeisel’s study, cells from dorsal midbrain were sequenced without accurate regional dissection of the superior colliculus. To identify SC cells, the authors computed regional enrichment for each cluster of the dorsal midbrain by correlating volumetric and RNA-seq gene expression profile. To compare the SC neuronal subtypes in our study to those in Zeisel’s study, we extracted excitatory neurons and inhibitory neurons from Zeisel’s work with the assigned regional identities of superior colliculus. In their study, SC excitatory neurons and inhibitory neurons were clustered into 4 (MEGLU1, MEGLU4, MELU5 and MEGLU6) and 6 clusters (MEINH5, MEINH6, MEINH7, MEINH8, MEINH10, MEINH11 and MEINH12) (Response Figure 1-figure supplement 2A-B), respectively, while 9 excitatory neuron (Ex-1 to Ex-9) and 10 inhibitory neuron (In-1 to In-10) clusters were identified in our study. We have shown more SC neuron subtypes than their work. To study the correspondences between neuronal subtypes reported in these two studies, we performed integration analysis on these two datasets followed the integration pipeline introduced by R package Seurat for excitatory neurons and inhibitory neurons independently (Figure 1-figure supplement 2C-F).

For excitatory neurons, we found MEGLU5 with assigned location of SuG layer (Zeisel et al. (2018)) were colocalized with Ex-8 and Ex-9 (this study) which have shown significant higher SuG layer specificity. MEGLU6 with Op identity was mapped onto Ex-6 which has been assigned to Op layer. MEGLU4 and Ex-7 both with SuG identities were highly similar to each other (Figure 1-figure supplement 2D). Other than that, MEGLU1, which was assigned to SC in general, was mapped onto clusters Ex-1 and Ex-2 (Figure 1-figure supplement 2D). In our study, by SPACED, Ex-1 and Ex-2 were labeled as InG/InWh and SC in general, respectively.

For inhibitory neurons, colocalization of cells with identical putative regional identities was also observed. Namely, MEINH10 and MEINH11 assigned to SuG layer were mapped with In-10 which had a regional identity of SuG. However, most subtypes (5 out of 7, including MEINH5/6/7/8/12) identified by Zeisel et al. were not assigned to any specific SC layers (Figure 1-figure supplement 2F). However, comparing to our data which with more clear spatial information, we revealed mappings between MEINH12 (Zeisel et al. (2018)) and In-4 (InG layer in this study), MEINH7 and In-2 (SuG layer in this study), MEINH6 and In-6 (SC in general in this study), MEINH5 and In-7 (SuG layer in this study), and MEINH8 and In-3 (Op layer in this study) (Figure 1-figure supplement 2F).

Overall, the integration analysis has shown good correspondences between neuronal subtypes identified in two studies. Plus, our SPACED methods offered more detailed spatial information of SC neural subtypes. These results were added in the revised manuscript as Figure 1-figure supplement 2 and also described in the text.

1.3. How does SPACED compare to previous methods for mapping single-cell transcriptomes on the Allen Brain Atlas data? For example, are the results of mapping with SPACED consistent with the results obtained with the mapping method proposed by Zeisel et al. 2018?

We thank the reviewer for the comments.

To compare the mapping results obtained from SPACED to that from the method developed by Zeisel et al., we computed layer specificity score for each excitatory neuron subtypes from SC dataset sequenced by Zeisel et al. using method SPACED (Author response image 2). As shown in Figure 1-figure supplement 2D, a subcluster of cells from MEGLU1 was mapped to Ex-1, which was assigned to InG layer in our study, indicating that spatial localization of those cells could be further revealed. Therefore, before performing SPACED, we subclustered excitatory neuron cluster MEGLU1 into four subclusters (MEGLU1-1 to MEGLU1-4) (Author response image 2). We then performed spatial mapping method SPACED based on the expression of ten most specific genes for each subcluster. As results, clusters MEGLU4 and MEGLU5 were assigned to SuG layer, while MEGLU6 exhibited high spatial score for the Op layer (Author response image 2). MEGLU1 subclusters MEGLU1-1, MEGLU1-3 and MEGLU1-4 didn’t show significant high layer specificity to any layers in the SC and therefore they were not assigned to any layers. The mapping results for those clusters were consistent with that revealed by Zeisel’s methods. Beyond that, MEGLU1 subcluster MEGLU1-2 was assigned to InG layer by SPACED which was not assigned to any specific SC layers by Zeisel’s methods. In summary, the mapping results obtained by SPACED are highly consistent with that from the method implemented by Zeisel et al. and SPACED has shown higher sensitivity for spatial mapping.

**Author response image 2. sa2fig2:** Comparison of mapping results for SC. (**A**) tSNE displaying the spatial distribution of excitatory neuron subtypes from Zeisel’s study. (**B**) Heatmap showing the layer specificity for each subtype computed by SPACED.

How does the method perform if the number of genes selected for mapping changes? How was the method benchmarked?

To test how the number of selected genes affects the mapping results of SPACED, we performed SPACED for excitatory neuron subtypes with 10, 15 and 20 DEGs per subtype independently. As shown in Author response image 3, with the increasing of selected genes, the number of assigned clusters decreased. Since for gene selection, we computed gene expression specificity among clusters and ranked genes in descending order based on their specificity value. Then for each cluster, the most-cluster-specific genes were selected for downstream SPACED analysis. Therefore, as the number of the genes included for spatial mapping increasing, the gene expression specificity decreases and may affect the results of spatial assignments. For reference, we have added the description of the impact of number of selected genes on the mapping results to our GitHub repository and recommended for using the top 10 most-cluster-specific genes for analysis (https://github.com/xiaoqunwanglab/SPACED) and also in the revised methods.

**Author response image 3. sa2fig3:** Impact of the number of selected genes on mapping results of SPACED. (A) Scatter plot showing the relation between number of selected genes and the number of assigned clusters. (B) Heatmap showing the computed layer specificity for each excitatory neuron subtype with 10 DEGs (left), 15 DEGs (middle) and 20 DEGs (right). * p < 0.05.

As we described above, SPACED achieved comparable results for mapping SC neurons to spatial locations and showed relative higher sensitivity than the method implemented by Zeisel et al.. For benchmarking, we then tested if SPACED is applicable for other brain regions. To this end, we applied SPACED on cortical pyramidal neurons sequenced by Zeisel et al. and compare the mapping results to that obtained in their study. As shown in Author response image 4, the cortical pyramidal neurons were grouped into 5 clusters and were assigned to 5 different cortical layers by Zeisel’s method. Using our SPACED method, the same mapping results were achieved (Author response image 4), as TEGLU7, TEGLU8, TEGLU10, TEGLU3 and TEGLU2 were assigned to cortical layer 2/3, layer4, layer5, layer6 and layer 6b, respectively. Therefore, besides SC, SPACED is also applicable for other brain regions, such as cortex and produced comparable mapping results. However, SPACED also has its limitation. As for in-situ hybridization image analysis, an important step of SPACED, we have to draw the ROIs (regions of interest) of the brain areas and then calculate signal intensity of genes and the region specificity sore as described in the Methods section in the manuscript. Therefore, SPACED is restricted to the brain regions which have clear anatomic structures and distinguishable borders, such as SC and cortex. For reference, we have added the description of methods limitation to our GitHub repository (https://github.com/xiaoqunwang-lab/SPACED).

**Author response image 4. sa2fig4:** Mapping cortical pyramidal neurons to spatial locations. (A) UMAP showing the distribution of cortical pyramidal neuron subtypes (left) and their probable locations (right). (B) Heatmap showing the layer specificity of each subtype revealed by SPACED.

2. Morphological reconstructions and Patch-Seq2.1. For the data shown in Figure 2A-G, the text refers to the injection of an unspecified "AAV mixture". Please describe in the main text what mixture was injected and what is the rationale of the experiment

We thank the reviewer for reminding us of clarifying the AAV mixture and the rationale of the sparse labeling experiment. The AAV mixture was made by mixing AAVTRE-DIO-Flpo (1x10^10^ particles/ml) and AAV-TRE-fDIO-GFP-IRES-tTA (1x10^12^ particles/ml) with a ratio of 1:9. This strategy took advantage of tTA-mediated leakage expression of GFP and have been shown to be efficient for sparse labeling of small number of neurons in specific brain areas (Lin et al., 2018). These contents have been added to the revised manuscript in Results and in Methods .

2.2. The integration of molecular and projection data relies entirely on the Patch-seq experiments. Figure 2J, however, shows that there are neurons retrogradely labeled from InG-ZI and Op-LPTN that do not express neither Cbln2 not Pitx2. Is it possible that these neurons belong to another excitatory type? In these experiments, retrograde tracing from LPTN and ZI is not cell type specific, therefore the existence of an excitatory type that projects to these two brain areas and does not express neither Cbln2 nor Pitx2 cannot be excluded. I suggest performing FISH for Cbln2 and Pitx2 on vGlut-IRES-Cre animals injected with AAV2-retro-DIO-EGFP in the LPTN or ZI.

We thank the reviewer for pointing this out. We agree with the reviewer that there are several cells express neither Cbln2 nor Pitx2 (12 out of 60; 6 from Op and 6 from InG). When we carried out integration analyses (Figure 2-figure supplement 1J), we found among these 12 cells, 6 cells mapped to Ex-2 (no layer specificity), 2 cells mapped to Ex-1 (considered as InG layer neurons), 2 cells mapped to Ex-8 (considered as SuG layer neurons), 1 cell mapped to Ex-9 (considered as SuG layer neurons) and 1 cell couldn’t map to any ExN clusters. These data indicate that the LPTN- or ZI- projecting neurons are diverse. Additionally, from retrograde tracing analysis, we also agree with the reviewer that there exist Cbln2-negative LPTN-projecting SC neurons and Pitx2-negative ZI-projecting SC neurons. Thus, in the revised manuscript, we do not claim that the Cbln2+ neurons are the only neuronal subtype that project to the LPTN. Similarly, we do not claim that the Pitx2+ neurons are the only neuronal subtype to project to the ZI.

2.3. Related to the previous point, the Methods mention that the Patch-seq data and the scRNAseq data are integrated using the CCA method in Seurat. However, here only a correlation matrix is shown (Figure 2K). A tSNE or UMAP showing the integrated scRNAseq and Patch-seq data would be much more informative.

We thank the reviewer for the suggestion. Figure 2K and Figure 2-figure supplement 1J shows the integration of snRNA-seq and patch-seq data by UMAP.

3. Anterograde tracing3.1 I would consider describing the experiments in Figure 6 earlier in the manuscript, after Figure 2. That seems the most logical place. Furthermore, figure S6 deserves to be part of one of the main figures. These are beautiful experiments showing very clearly that Cbln2 and Pitx2 neurons project to different brain areas. However, the narrative of the paper is prone to mislead the reader, because it emphasizes the projections to LPTN and ZI.

We thank the reviewer for these great suggestions. We feel that the characterizations of Cbln2-IRES-Cre and Pitx2-Cre lines to specifically label Cbln2+ and Pitx2+ SC neurons (Figure 3A-B,H-I) had better go first before showing the downstream target regions of these neurons. Thus, we would politely keep Figure 6 in its original slot. However, we do accept the reviewer’s second suggestion that Figure S6 in the original manuscript deserves to be part of main figures. Figure S6 in the original manuscript has been moved to follow Figure 5 in the revised manuscript.

3.2 Related to the previous point, it would be helpful to expand the Discussion by mentioning how the different brain areas innervated by Cbln2 and Pitx2 neurons may participate to predator escape and prey capture behaviors.

We thank the reviewer for this good suggestion. In the revised manuscript, we have expanded the Discussion about how the different target areas of Cbln2+ and Pitx2+ neurons may participate in predator avoidance and prey capture behaviors. In addition, we have added three new references to support these discussions.

4. Calcium imagingIt is good to see representative traces from the fiber photometry data. However, it would be even nicer to show a raster plot or a heat map encompassing all the neurons that were recorded.

We thank the reviewer for this great suggestion. In Figure 4 of the revised manuscript, we have shown heat maps of individual trials of the example Cbln2-IRES-Cre and Pitx2-Cre mice that were used for fiber photometry recording.

I could not find anywhere in the manuscript how many neurons were recorded and from how many animals. Information on the data analysis pipeline for these data is also missing (or at least I could not find it easily). All this information should be included in the manuscript.

In Figure 4 of the revised manuscript, we have shown the number of mice used for the analyses in Figure 4H-K. In addition, the animal number we used for each experiment was shown in Supplementary File 8.